

# Assessing the impacts of 1.5°C global warming – simulation protocol of the Inter-Sectoral Impact Model Intercomparison Project (ISIMIP2b)

Katja Frieler[1], Richard Betts[2], Eleanor Burke[2], Philippe Ciais[3], Sebastien Denvil[4], Delphine Deryng[5], Kristie Ebi[6], Tyler Eddy[7,8], Kerry Emanuel[9], Joshua Elliott[5,10], Eric Galbraith[11], Simon N. Gosling[12], Kate Halladay[2], Fred Hattermann[1], Thomas Hickler[13], Jochen Hinkel[14,15], Veronika Huber[1], Chris Jones[2], Valentina Krysanova[1], Stefan Lange[1], Heike K. Lotze[7], Hermann Lotze-Campen[1,16], Matthias Mengel[1], Ioanna Mouratiadou[1,17], Hannes Müller Schmied[13,18], Sebastian Ostberg[1,23], Franziska Piontek[1], Alexander Popp[1], Christopher P.O. Reyer[1], Jacob Schewe[1], Miodrag Stevanovic[1], Tatsuo Suzuki[19], Kirsten Thonicke[1], Hanqin Tian[20], Derek P. Tittensor[7,21], Robert Vautard[3], Michelle van Vliet[22], Lila Warszawski[1], Fang Zhao[1]

[1] Potsdam Institute for Climate Impact Research, Potsdam, 14473, Germany
[2] Met Office, Exeter, UK
[3] Laboratoire des Sciences du Climat et de l'Environnement, Gif sur Yvette, France
[4] Institut Pierre-Simon Laplace, Paris, France

[5] NASA GISS/CCSR Earth Institute, Columbia University, New York, NY, USA
[6] University of Washington, Seattle, WA, USA
[7] Department of Biology, Dalhousie University, Halifax, Nova Scotia, Canada
[8] Institute for Oceans and Fisheries, University of British Columbia, Vancouver, British Columbia, Canada
[9] Program for Atmospheres, Oceans and Climate, Massachusetts Institute of Technology, Cambridge, Massachusetts
[10] Computation Institute, University of Chicago, Chicago, IL, USA
[11] Catalan Institution for Research and Advanced Studies, Barcelona, Spain
[12] School of Geography, University of Nottingham, United Kingdom

[13] Senckenberg Biodiversity and Climate Research Centre (BiK-F), Frankfurt, Germany
[14] Global Climate Forum, 10178 Berlin, Germany
[15] Division of Resource Economics, Albrecht Daniel Thaer-Institute and Berlin Workshop in Institutional Analysis of Social-Ecological Systems (WINS), Humboldt-University, Berlin, Germany
[16] Humboldt-Universität zu Berlin, Berlin, Germany
[17] Copernicus Institute of Sustainable Development, Utrecht University, Utrecht, The Netherlands
[18] Institute of Physical Geography, Goethe-University Frankfurt, Germany
[19] Japan Agency for Marine-Earth Science and Technology, Department of Integrated Climate Change Projection Research, Yokohama, Japan
[20] International Center for Climate and Global Change Research, School of Forestry and Wildlife Sciences, Auburn University, Auburn, AL, USA
[21] United Nations Environment Programme World Conservation Monitoring Centre, Cambridge, UK
[22] Water Systems and Global Change group, Wageningen University, Wageningen, The Netherlands
[23] Geography Department, Humboldt-Universität zu Berlin, Berlin, Germany





*Correspondence to*: Katja Frieler (katja.frieler@pik-potsdam.de)

**Abstract.** In Paris, France, December 2015, the Conference of the Parties (COP) to the United Nations Framework Convention on Climate Change (UNFCCC) invited the Intergovernmental Panel on Climate Change (IPCC) to provide a "special report in 2018 on the impacts of global warming of 1.5°C above pre-industrial levels and related global greenhouse gas emission pathways". In Nairobi, Kenya, April 2016, the IPCC panel accepted the invitation. Here we describe the response devised within the Inter-Sectoral Impact Model Intercomparison Project (ISIMIP) to provide tailored, cross-sectorally consistent impacts projections. The simulation protocol is designed to allow for 1) separation of the impacts of historical warming starting from pre-industrial conditions from other human drivers such as historical land-use changes (based on pre-industrial and historical impact model simulations); 2) quantification of the effects of additional warming up to 1.5°C, including a potential overshoot and long-term effects up to 2299, compared to a no-mitigation scenario (based on the low-emissions Representative Concentration Pathway RCP2.6 and a no-mitigation pathway RCP6.0) with socio-economic conditions fixed at 2005 levels; and 3) assessment of the climate effects based on the same climate scenarios but accounting for simultaneous changes in socio-economic conditions following the middle-of-the-road Shared Socioeconomic Pathway (SSP2, Fricko et al., 2016) and differential bio-energy requirements associated with the transformation of the energy system to comply with RCP2.6 compared to RCP6.0. With the aim of providing the scientific basis for an aggregation of impacts across sectors and analysis of cross-sectoral interactions that may dampen or amplify sectoral impacts, the protocol is designed to facilitate consistent impacts projections from a range of impact models across different sectors (global and regional hydrology, global crops, global vegetation, regional forests, global and regional marine ecosystems and fisheries, global and regional coastal infrastructure, energy supply and demand, health, and tropical cyclones).

## 1    Introduction

Societies are strongly influenced by weather and climate conditions. On the one hand, persistent weather patterns influence lifestyle, infrastructures, and agricultural practices across climatic zones. On the other hand, individual weather events have the potential to cause direct damages. However, the translation of projected changes in weather and climate into impacts on the functioning of societies will manifest through a complex system of pathways which is not yet fully understood and captured by comprehensive predictive models (Warren, 2011). Direct economic losses from weather extremes represent only a fraction of the total impacts that could be expected, yet even this subset, which is relatively straightforward to model, amounts to about $US95 billion per





year on average over 1980-2014 (Munich Re, 2015). In addition to these purely economic effects, from 2008 to 2015 an estimated 21.5 million people per year were displaced by weather events (Internal Displacement Monitoring Centre and Norwegian Refugee Council, 2015). The causes are diverse: storms accounted for 51% of the physical damages of weather events, flood and mass movements induced 32%, and extreme temperatures,

droughts and wildfire inflicted 17% of the economic damages. Displacement was mainly driven by floods (64%) and storms (35%), with minor contributions from extreme temperatures (0.6%), wet mass movement (0.4%), and wildfires (0.2%). The more indirect effects of rainfall deficits and agricultural droughts on displacement are not captured in these global statistics of displacement. Even very basic projections of fluctuations and long-term trends in the underlying drivers of direct economic damages or displacement require a range of different types of

climate impacts models (e.g. hydrological models for flood risks, biomes models for risks of wildfires, crop models for heat or drought-induced crop failure), which have to be forced by the same climate input to allow for an aggregation of the respective impacts.

Providing these consistent impacts projections is critical since global warming will drive changes in both the

average weather patterns and the spatial distribution, intensity and frequency of extreme events such as droughts, high precipitation, heat waves and storms. Regional trends (Hartmann et al., 2013) and changes in heat-wave frequency (Barriopedro et al., 2011) and precipitation records (Lehmann et al., 2015) are already emerging and are expected to strengthen with increasing global warming. The shear diversity of mechanisms by which these changes will affect our societies seriously limits our ability to project the societal impacts of climate change.

In particular, other socio-economic changes are expected to increase or decrease impacts by altering underlying patterns of vulnerability and exposure. This complexity problem has led to the development of empirical approaches, linking pure climate indicators like temperature or precipitation to highly-aggregated socio-economic indicators such as national Gross Domestic Product (GDP), without resolving the underlying mechanisms (Burke et al., 2015; Dell et al., 2012). However, the plausibility of the extrapolation of the historical temperature sensitivity

used by empirical models should be evaluated based on process-based impact projections, which provide a more detailed implementation of the underlying drivers. If this is not done, the evaluation may fail to identify underlying interactions and non-linear effects.

Despite the lack of comprehensive, integrated, global impact models, a growing array of focused impact models

translate projected changes in climate and weather into changes in individual sectors, including vegetation cover, crop yields, marine ecosystems and fishing potentials, frequency and intensity of river floods, coastal flooding due to sea level rise, water scarcity, distribution of vector-borne diseases, heat and cold-related mortality, labour



productivity, and energy supply (e.g. hydropower potentials) and demand. With these models, we can move beyond quantification of climate change alone towards a more process-based quantification of societal risks. For example, this can be done by developing empirical approaches using not only climate indicators such as temperature, but also impact indicators, such as losses in crop production or labour productivity, as predictors for

changes in socio-economic development. Although the sector-specific impact models are constructed independently and do not interact (except for a few multi-sector models), by considering their behaviour within a single simulation framework, we can start to understand the true, integrated impacts of climate change. This undertaking demands a framework for coordinating and analysing sector-specific models, analogous to the physical climate-modelling community's Coupled Model Intercomparison Project (CMIP).

ISIMIP is designed to address this challenge by forcing a wide range of climate-impact models with the same climate and socio-economic input (Schellnhuber et al., 2013, www.isimip.org). Its first phase (the Fast Track) provided the first set of cross-sectorally consistent, multi-model impact projections (Warszawski et al., 2014). The data are publicly available through https://esg.pik-potsdam.de. Meanwhile the project is in its second phase, in

which the first simulation round (ISIMIP2a) was dedicated to historical simulations with a view to detailed model evaluation, in particular with respect to the impacts of extreme events. So far, around 60 international modelling groups have submitted data to the ISIMIP2a repository, which will eventually be made publicly available. Here, we describe the simulation protocol and scientific rationale for the next round of simulations (ISIMIP2b). The protocol was developed in response to the planned IPCC Special Report on the 1.5°C target, reflecting the

responsibility of the impacts-modelling community to provide the best scientific basis for political discussions about mitigation and adaptation measures. Importantly, the simulations also offer a broad basis for climate impacts research beyond the scope and time frame of the Special Report. Therefore, similarly to the climate simulations generated within the Coupled Model Intercomparison Project (CMIP, Taylor et al., 2012), the ISIMIP simulation data will be made publicly available to support further studies of climate-change impacts relative to

pre-industrial levels.

In Paris, parties agreed on "…holding the increase in the global average temperature to well below 2 °C above pre-industrial levels and pursuing efforts to limit the temperature increase to 1.5°C above pre-industrial levels, recognizing that this would significantly reduce the risks and impacts of climate change." (UNFCCC, 2015). While

the statement "holding below 2°C" implies keeping global warming below the 2°C limit over the full course of the century and afterwards, "efforts to limit the temperature increase to 1.5°C" is often interpreted as allowing for a potential overshoot before returning to below 1.5°C (Rogelj et al., 2015). Given the remaining degrees of freedom





regarding the timing of maximum warming and the length of an overshoot, the translation of emissions into global mean temperature change, and, even more importantly, the uncertainty in associated regional climate changes, a wide range of climate change scenarios, all consistent with these political targets, should be considered. However, the computational expense of climate and climate-impact projections limits the set of

scenarios that can be feasibly computed. These should therefore be carefully selected to serve as the basis for efficient extrapolations of impacts to a wider range of relevant climate-change scenarios. In the ISIMIP2b protocol, the Representative Concentration Pathway (RCP) RCP2.6 was chosen, being the lowest emission scenario considered within CMIP5 and in line with a 1.5°C or 2°C limit of global warming depending on the definition and the considered Global Circulation Model (GCM). While there are plans within the next phase of

CMIP to generate climate projections for a lower emission scenario (RCP2.0), these data will not be available in time to do the associated impacts projections for the Special Report.

The ISIMIP protocol covers a core set of scenarios that can be run by all participating impact-modelling groups, ensuring a minimal set of multi-model impact simulations consistent across sectors, and therefore allowing for

cross-sectoral aggregation and integration of impacts. Core simulations will focus on 1) quantification of impacts of the historical warming compared to pre-industrial reference levels (see Figure 1a, Group 1), 2) quantification of the climate change effects based on RCP2.6 and RCP6.0 assuming fixed, present-day management, land-use (LU) and irrigation patterns and societal conditions (see Figure 1a, Group 2) including a quantification of the long-term effects of low-level global warming following a potential overshoot based on an extended RCP2.6 simulations to

2299, and 4) quantification of the impacts of "low-level" (~1.5°C) global warming based on RCP2.6 compared to RCP6.0, while accounting for additional human influences such as changes in management and LU patterns in response to population growth and bioenergy demand (see Figure 1b, Group 3). Since comprehensive cross-sectoral aggregation and integration of impacts is still in its early stages, ISIMIP2b is not intended to capture the full spread of impact projections induced by the uncertainty in climate projections or the range of different socio-

economic-development pathways. Rather, it provides the framework for a set of exemplary model simulations that can be used to develop the tools required for a better representation of interactions of climate-change impacts or mitigation measures (see Figure 1b, Group 3) across sectors. To this end, simulations are prioritized to ensure the generation of such a data set of consistent climate impacts projections for at least one exemplary set of climate and socio-economic input data, even if modelling groups may not have the capacity to provide all

simulations covered by the protocol.



In section 2 of the paper we outline the basic set of scenarios and the rationale for their selection. Sections 3-6 provide a more detailed description of the input data represented in Figure 1, i.e. climate input data, LU and irrigation patterns accounting for mitigation-related expansion of managed land (e.g. for bioenergy production), population and GDP data, and associated harmonized input representing "other human influences". Section 7

provides some information about the sector-specific implementation of the different scenarios while the list of requested output variables is part of the supporting online material (SOM). It is important to note that this paper describes the scientific rationale and key scenario characteristics - up-to-date and detailed instructions, such as e.g. conventions on the file names and formats, is to be found in the ISIMIP2b simulation protocol on the ISIMIP website (www.isimip.org). Modellers should always refer to this online document when setting up and

performing simulations.

## 2    The rationale of the basic scenario design

To ensure wide sectoral coverage by a large number of impact models, the set of scenarios is restricted to 1) one future socio-economic pathway (SSP2 representing middle-of-the-road socio-economic development concerning population and mitigation and adaptation challenges (O'Neill et al., 2014), see section 5); 2) climate input from

four global climate models (GCMs) (for all of which pre-industrial control simulations are available), 3) simulations of the historical period, and future projections for the no-mitigation baseline scenario for the SSP2 storyline (SSP2 + RCP6.0) (Fricko et al., 2016) and the strong mitigation scenario (SSP2 + RCP2.6) closest to the global warming limits agreed on in Paris (see section 3); and 4) representation of potential LU and irrigation changes associated with SSP2 + RCP6.0 and SSP2 + RCP2.6 as generated by the global LU model MAgPIE (Model of Agricultural

Production and its Impact on the Environment) (Lotze-Campen et al., 2008; Popp et al., 2014a; Stevanović et al., 2016), which account for climate-induced changes in crop production, water availability and terrestrial carbon content as projected by the LPJmL model (Bondeau et al., 2007) and differential bio-energy application (see section 4). Land-based mitigation for MAgPIE is driven by carbon prices and bioenergy demand generated within the Integrated Assessment Modelling Framework REMIND-MAgPIE, as implemented in the SSP exercise (Kriegler

et al., 2016). The native data on LU and irrigation changes from these REMIND-MAgPIE SSP runs do not account for the impacts of climate change and increasing atmospheric $CO_2$ concentrations on crop production, water availability and terrestrial carbon content (Popp et al., 2016) and therefore differ from the patterns to be used within ISIMIP2b.



## 2.1 Quantification of pure climate-change effects of the historical warming compared to pre-industrial reference levels (Figure 1a, Group 1)

The Paris agreement explicitly asks for an assessment of "the impacts of global warming of 1.5°C above pre-industrial levels". Usually, impact projections (such as those generated within the ISIMIP Fast Track, Warszawski

et al., 2013) only allow for a quantification of projected impacts (of say 1.5°C warming) compared to "present day" or "recent past" reference levels, because the impacts model simulations rarely cover the pre-industrial period. Despite this, the "detection and attribution" of historical impacts is a highly relevant field of research, in particular in the context of the "loss and damage" debate (James et al. 2014), but also to better understand the processes affecting the climate-change impacts already unfolding today. In the Fifth Assessment Report of the

Intergovernmental Panel on Climate Change (IPCC AR5), one chapter is dedicated to the detection and attribution of observed climate-change impacts (Cramer et al., 2014). However, the conclusions that can be drawn are limited by: 1) the lack of long and homogeneous observational data, and 2) the confounding influence of other drivers such as population growth and management changes (e.g. expansion of agriculture in response to growing food demand, changes in irrigation water withdrawal, building of dams and reservoirs, changes in fertilizer input,

and switching to other varieties) on climate impact indicators such as river discharge, crop yields and energy demand. Over the historical period, these human influences have evolved simultaneously with climate, rendering the quantification of the pure climate-change signal difficult. Model simulations could help to fill these gaps and could become essential tools to separate the effects of climate change from other historical drivers.

To address these challenges the ISIMIP2b protocol includes: 1) a multi-centennial pre-industrial reference simulation (picontrol + constant pre-industrial levels of other human influences (1860soc), 1660-1860); 2) historical simulations accounting for varying human influences but assuming pre-industrial climate (picontrol + histsoc, 1861-2005); 3) historical impact simulations accounting for varying human influences and climate change (historical + histsoc, 1861-2005). These scenarios facilitate the separation of the effects of historical warming (as

simulated by GCMs) from other human influences by taking the difference between the two model runs covering the historical period. The full period of historical simulation results also allows for cross-sectorial assessments of when the climate signal becomes significant. In addition, the control simulations will provide a large sample of pre-industrial reference conditions allowing for robust determination of extreme-value statistics (e.g. extreme events such as one hundred year flood events) and e.g. the typical spatial distribution of impacts associated with

certain large-scale circulation patterns such as El Nino (Iizumi et al., 2014; Ward et al., 2014) or other circulation regimes capable of synchronising the occurrence of extreme events across sectors and regions (Coumou et al., 2014; Francis and Vavrus, 2012). In addition, the pre-industrial reference represents a more realistic starting point



(and spin-up conditions) for e.g. the vegetation models or marine ecosystem models compared to starting from artificial "equilibrium present day" conditions as was done in the ISIMIP fast track.

For models that are not designed to represent temporal changes in LU patterns or "other human influences"
simulations should be based on constant present day (year 2005) societal conditions ("2005soc", dashed line in Figure 1a). Modelling teams whose models do not account for any human influences are also invited to contribute simulations for Group 1 and Group 2 based on naturalized settings (to be labelled "nosoc"). A detailed documentation of the individual model-specific settings implemented by the different modelling groups will be made accessible on the ISIMIP website.

**2.2   Future impact projections accounting for low (RCP2.6) and high (RCP6.0) Greenhouse gas emissions assuming present day socio-economic conditions (Figure 1a, Group 2)**

To quantify the pure effect of additional warming to 1.5°C above pre-industrial levels or higher, the scenario choice includes a group of future projections assuming other human influences fixed at present day (chosen to be 2005) conditions (2005soc, see Figure 1, Group 2). The Group 2 simulations start from the Group 1 simulations
and assume: 1) fixed, year 2005 levels of human influences but pre-industrial climate (picontrol + 2005soc, 2006-2100), 2) fixed year 2005 levels of human influences and climate change under the strong-mitigation scenario RCP2.6 (rcp26 + 2005soc, 2006-2100), 3) fixed year 2005 levels of human influences and climate change under the no-mitigation scenario RCP6.0 (rcp60 + 2005soc, 2006-2100), and 4) extension of the RCP2.6 simulations to 2299 assuming human influences fixed at year 2005 levels (rcp26 + 2005soc, 2101-2299). In this way, the distribution of
impact indicators within certain time windows, in which global warming is around e.g. 1.5°C or 2°C, can be compared without the confounding effects of other human drivers that vary with time (e.g. Fischer and Knutti, 2015; Schleussner et al., 2015). In particular, the impacts at these future levels of warming can be compared to the pre-industrial reference climate, assuming a representation of pre-industrial levels of socio-economic conditions (picontrol + 1860soc, Group 1) and pre-industrial reference climate but present-day levels of other
human influences (picontrol + 2005soc, Group 2).

The extension of the RCP2.6 projections to 2299 is critical because: 1) global mean temperature may only return to warming levels below 2°C after 2100 (see HadGEM2-ES and IPSL-CM5A-LR, Figure 2), and 2) impacts of global warming will not necessarily emerge in parallel with global mean temperature change, because, for example,
climate models show a hysteresis in the response of the hydrological cycle (Wu et al., 2010) due to ocean inertia. Similarly, sea-level rise associated with a certain level of global warming will only fully manifest over millennia. In





addition to the lagged responses of the forcing data to Greenhouse gas emissions, there is inertia in the affected systems (such as vegetation changes and permafrost thawing) that will delay responses. Thus, an assessment of the risks associated with 1.5°C global warming requires simulations of impacts when 1.5°C global warming is reached, as well as of the impacts when global warming returns to 1.5°C and stabilizes. Given that RCP2.6 may

exceed 1.5°C warming (depending on the exact definition of the temperature goal and the applied climate model) the characteristic peak and decline in global mean temperature will help to get a better understanding of the associated impacts dynamics. This could be used to derive reduced-form approximations of the complex-model simulations, allowing for a scaling of the impacts to other global-mean-temperature and $CO_2$ pathways. Providing the basis for the development of these tools is critical given the range of scenarios consistent with the

temperature goals as described in the Paris agreement.

Depending on the time scale of stabilization of the climate and the lag in the response of the impacts to climate change the extension of the simulations to 2299 could provide a sample of a relatively stable distribution of impacts associated with RCP2.6 levels of emissions. Similar to the 200-year pre-industrial reference simulations,

this sample could provide a basis for the estimation of extreme-value distributions that can be compared to the associated pre-industrial reference distributions (picontrol + 1860soc (Group 1) or picontrol + 2005soc (Group 2)).

### 2.3  Future impact projections accounting for low (RCP2.6) and high (RCP6.0) levels of climate change accounting for socioeconomic changes (Figure 1b, Group 3)

Future projections of the impacts of climate change will also depend on future socio-economic development, just

like historic impacts. For example many impact indicators such as "number of people affected by flood events" (Hirabayashi et al., 2013) or "number of people affected by long-term changes going beyond a certain range of the reference distribution" (Piontek et al., 2014) directly depend on population projections (exposure) or socio-economic conditions e.g. as reflected in flood protection levels (vulnerability). While socio-economic drivers can partly be accounted for in post-processing (e.g. for the number of people affected by tropical cyclones) others are

directly represented in the models such as dams and reservoirs or LU changes. To capture the associated effects on the impact indicators, the ISIMIP2b protocol also contains a set of future projections accounting for potential changes in socio-economic conditions (e.g. rcp26soc), building on the SSP2 story line (see Figure 1, Group 3). The relevance and representation of specific socio-economic drivers strongly differs from sector to sector or impact model to impact model. Here, we focus on changes 1) in population patterns and national GDP (see section 6), 2)

land-use and irrigation patterns (see section 4), and 3) fertilizer input, and nitrogen deposition (see section 7). However, even beyond these indicators, models that represent other individual drivers should account for



associated changes according to their own implementation of the SSP2 storyline. The simulations start from the Group 1 simulations and assume 1) future changes in human influences but pre-industrial climate (picontrol + rpc26soc/rcp60soc, 2006-2100), 2) future changes in human influences and climate change under the strong mitigation scenario RCP2.6 (rcp26 + rcp26soc, 2006-2100), 3) future changes in human influences and climate

change under the no-mitigation scenario RCP6.0 (rcp60 + rcp60soc, 2006-2100), and 4) and extension of the RCP2.6 simulations to 2299 assuming human influences fixed at 2100 levels (rcp26 + 2100rcp26soc, 2101-2299).

The representation of LU and irrigation change is particularly challenging as it will not only reflect the potential expansion of agricultural land due to increasing demand induced by 1) population growth and 2) changing diets

under economic development, but also due to 3) climate-change effects on crop yields, and 4) bioenergy demand associated with the level of climate change mitigation. The ISIMIP2b protocol is designed to account for all four aspects (see section 4). Using associated LU patterns in the impact models participating in ISIMIP2b will allow for an exemplary assessment of potential side-effects of certain transformations of the energy system associated with a 1.5°C global-mean-temperature limit, such as the allocation of land areas to bioenergy production. The

scenario design will facilitate estimation of the consequences of the suggested LU changes in comparison to the avoided impacts of climate change.







**Figure 1** Schematic representation of the scenario design for ISIMIP2b. "Land use" also includes irrigation. "Other" includes other non-climatic anthropogenic forcing factors and management, such as fertilizer input, selection of crop varieties, flood protection levels, dams and reservoirs, water abstraction for human use, fishing effort, atmospheric nitrogen deposition, etc. Panel a) shows the Group 1 and Group 2 runs. Group 1 consists of model runs to separate the pure effect of the historical climate change from other human influences. Models that cannot account for changes in a particular forcing factor are asked



to hold that forcing factor at 2005 levels (2005soc, dashed lines). Group 2 consists of model runs to estimate the pure effect of the future climate change assuming fixed year 2005 levels of population, economic development, land use and management (2005soc). Panel b) shows Group 3 runs. Group 3 consists of model runs to quantify the effects of the land use (and irrigation) changes, and changes in population, GDP, and management from 2005 onwards associated with RCP6.0 (no mitigation scenario under SSP2) and RCP2.6 (strong mitigation scenario under SSP2). Forcing factors for which no future scenarios exist (e.g. dams/reservoirs) are held constant after 2005.

## 3    Climate input data

Bias-corrected climate input data at daily temporal and 0.5° horizontal resolution representing pre-industrial, historical and future (RCP2.6 and RCP6.0) conditions will be provided based on CMIP5 output of GFDL-ESM2M, HadGEM2-ES, IPSL-CM5A-LR and MIROC5. Output from the first three of these four GCMs was already used in the ISIMIP fast track. In contrast to the ISIMIP fast track we will also provide bias-corrected atmospheric data over the ocean, which is, for example, relevant for the impacts on offshore energy generation or the physical representation of coastal flooding. Output from two of the GCMs (GFDL-ESM2M and IPSL-CM5A-LR) includes the physical and biogeochemical ocean data required by the marine ecosystem sector of ISIMIP (see FISH-MIP, www.isimip.org/gettingstarted/marine-ecosystems-fisheries/). The fast-track model NorESM1-M was taken out of the selection due to the unavailability of near-surface wind data, and MIROC-ESM-CHEM was replaced by MIROC5, which in comparison features twice the horizontal atmospheric resolution (Watanabe et al., 2010, 2011), a lower equilibrium climate sensitivity (Flato et al., 2013), a smaller temperature drift in the pre-industrial control run (0.36°C/ka compared to 0.93°C/ka), and more realistic representations of ENSO (Bellenger et al., 2014), the Asian summer monsoon (Sperber et al., 2013) and North Atlantic extratropical cyclones (Zappa et al., 2013) during the historical period.

GCM selection was heavily constrained by CMIP5 data availability since we employed a strict climate input data policy to facilitate unrestricted cross-sectoral impact assessments. In order to be included in the selection, daily CMIP5 GCM output had to be available for the atmospheric variables listed in Table 1 covering at least 200 pre-industrial control years, the whole historical period from 1861 to 2005, and RCP2.6 and RCP6.0 from 2006 to 2099 each. Currently, these requirements are completely met for GFDL-ESM2M, IPSL-CM5A-LR and MIROC5. Existing gaps in HadGEM2-ES data (see Figure 2) will be filled by re-running the model accordingly, which has been kindly agreed to by the responsible modelling teams. A MIROC5 extension of RCP2.6 until 2299 will become available soon.

Impact-model simulations using climate input data from IPSL-CM5A-LR and GFDL-ESM2M are the first and second priority climate input data sets respectively, since these GCMs provide all the monthly ocean data required by



FISH-MIP and since IPSL-CM5A-LR additionally offers an extended RCP2.6 projection. Usage of MIROC5 data is of third priority. Since HadGEM2-ES climate input data will only become available in the course of the project it is the fourth priority.

5    Global-mean-temperature projections from IPSL-CM5A-LR and HadGEM2-ES under RCP2.6 exceed 1.5°C relative to pre-industrial levels in the second half of the 21st century (see Figure 2). While global-mean-temperature change returns to 1.5°C or even slightly lower by 2299 in HadGEM2-ES, it only reaches about 2°C in IPSL-CM5A-LR by 2299. For GFDL-ESM2M, global-mean-temperature change stays below 1.5°C until 2100. For MIROC5, it stabilizes at about 1.5°C during the second half of the 21st century.

For HadGEM2-ES, IPSL-CM5A-LR and MIROC5, it was necessary to recycle pre-industrial control climate data in order to fill the entire 1661–2299 period. Based on available data, the recycled time series start after the first 320 (HadGEM2-ES), 440 (IPSL-CM5A-LR) and 570 (MIROC5) pre-industrial control years, which means that pre-industrial control climate data from 1981, 2101 and 2231 onwards are identical to those from 1661 onwards,

15   respectively. For GFDL-ESM2M, no such recycling was necessary. For all four GCMs, temperature drifts in the pre-industrial control run are considered sufficiently small relative to inter-annual variability and temperature changes in the historical and future periods, so that de-trending pre-industrial control climate data was deemed unnecessary.



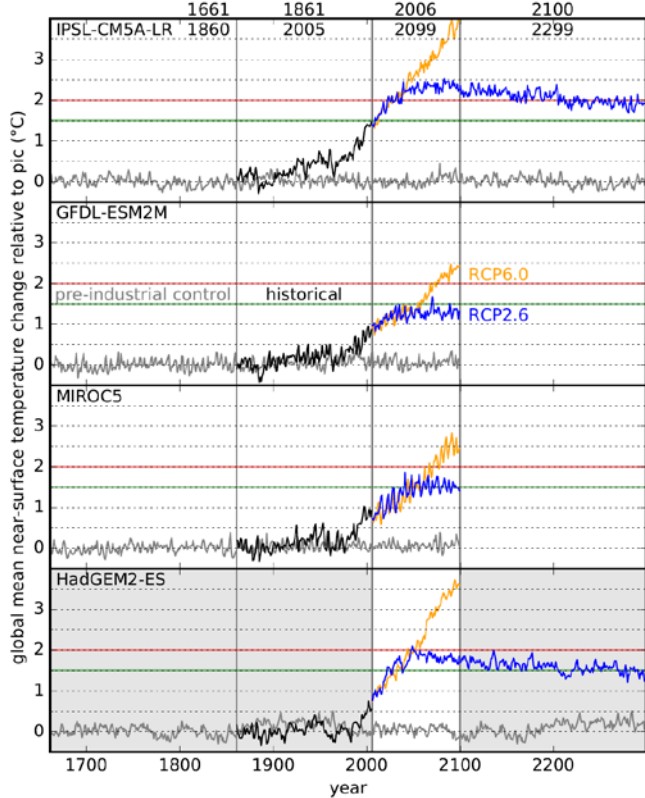

**Figure 2** Time series of annual global mean near-surface temperature change relative to pre-industrial levels (1361-1860) as simulated with IPSL-CM5A-LR, GFDL-ESM2M, MIROC5 and HadGEM2-ES (from top to bottom). Colour coding indicates the underlying CMIP5 experiments (grey: pre-industrial control, black: historical, blue: RCP2.6, yellow: RCP6.0) with corresponding time periods given at the top. Grey shading indicates model-experiment combinations with currently incomplete CMIP5 data.

For most variables, the provided atmospheric GCM data have been bias-corrected using slightly modified versions of the ISIMIP fast-track methods, which correct multi-year monthly mean values, such that trends are preserved in absolute and relative terms for temperature and non-negative variables respectively, and derives transfer functions to correct the distributions of daily anomalies from monthly mean values (Hempel et al., 2013). Known issues of the fast-track methods are: 1) humidity was not corrected since the methods were not designed for variables with both lower and upper bounds, such as relative humidity, and since their application to specific humidity yields relative humidity statistics that compare poorly with those observed; 2) bias-corrected daily mean shortwave radiation values too frequently exceed 500 W m$^{-2}$ over Antarctica and high-elevation sites; 3) for pressure, wind speed, longwave and shortwave radiation they produce discontinuous daily climatologies as described by Rust et al. (2015) for the WATCH forcing data (Weedon et al., 2011), 4) they occasionally generate spuriously high precipitation events in semi-arid regions, and 5) they do not adjust the interannual variability of



monthly mean values, which would be an important improvement for the purpose of impact projections (Sippel et al., 2016). While 5) and 4) are items of future work, problems 3), 2) and 1) were solved through modifcations of the correction methods for pressure, wind speed and longwave radiation (see below), and by using newly developed, approximately trend-preserving bias correction methods for relative humidity and shortwave

radiation (Lange et al., 2016a).

In addition to these adjustments, we correct to a new reference data set. While in the fast track, WATCH forcing data (Weedon et al., 2011) were employed for bias correction, the ISIMIP2b forcing data are corrected to the newly compiled reference dataset EWEMBI (E2OBS, WFDEI and ERAI data Merged and Bias-corrected for ISIMIP),

which covers the entire globe at 0.5° horizontal and daily temporal resolution from 1979 to 2013. Data sources of EWEMBI are ERA-Interim reanalysis data (ERAI; Dee et al., 2011), WATCH forcing data methodology applied to ERA-Interim reanalysis data (WFDEI; Weedon et al., 2014), eartH2Observe forcing data (E2OBS; Dutra, 2015) and NASA/GEWEX Surface Radiation Budget data (SRB; Stackhouse Jr. et al., 2011). The SRB data were used to bias-correct E2OBS short-wave and long-wave radiation using a new method that has been developed particularly for

this purpose (Lange et al., 2016b) in order to reduce known deviations of E2OBS radiation statistics from the respective SRB estimates over tropical land (Dutra, 2015). Data sources of individual EWEMBI variables are given in Table 1.



**Table 1** Data sources of individual variables of the EWEMBI dataset. Note that E2OBS data are identical to WFDEI over land and ERAI over the ocean, except for precipitation over the ocean, which was bias-corrected using GPCPv2.1 monthly precipitation totals (Balsamo et al., 2015; Dutra, 2015). WFDEI-GPCC means WFDEI with GPCCv5/v6 monthly precipitation totals used for bias correction (Weedon et al., 2014; note that the WFDEI precipitation products included in E2OBS were those that were bias-corrected with CRU TS3.101/TS3.21 monthly precipitation totals). E2OBS-SRB means E2OBS with SRB daily mean radiation used for bias correction (Lange et al., 2016b). E2OBS-ERAI means E2OBS everywhere except over Greenland and Iceland (cf. Weedon et al., 2010, p. 9), where monthly mean diurnal temperature ranges were restored to those of ERAI using the Sheffield et al. (2006) method. Note that precipitation here means total precipitation, i.e., rainfall plus snowfall.

| Variable | Short name | Unit | Source dataset over land | Source dataset over the ocean |
|---|---|---|---|---|
| Near-Surface Relative Humidity | hurs | % | E2OBS | E2OBS |
| Near-Surface Specific Humidity | huss | $kg\ kg^{-1}$ | E2OBS | E2OBS |
| Precipitation | pr | $kg\ m^{-2}\ s^{-1}$ | WFDEI-GPCC | E2OBS |
| Snowfall Flux | prsn | $kg\ m^{-2}\ s^{-1}$ | WFDEI-GPCC | E2OBS |
| Surface Air Pressure | ps | Pa | E2OBS | E2OBS |
| Surface Downwelling Longwave Radiation | rlds | $W\ m^{-2}$ | E2OBS-SRB | E2OBS-SRB |
| Surface Downwelling Shortwave Radiation | rsds | $W\ m^{-2}$ | E2OBS-SRB | E2OBS-SRB |
| Near-Surface Wind Speed | sfcWind | $m\ s^{-1}$ | E2OBS | E2OBS |
| Near-Surface Air Temperature | tas | K | E2OBS | E2OBS |
| Daily Maximum Near-Surface Air Temperature | tasmax | K | E2OBS-ERAI | E2OBS |
| Daily Minimum Near-Surface Air Temperature | tasmin | K | E2OBS-ERAI | E2OBS |
| Eastward Near-Surface Wind | uas | $m\ s^{-1}$ | ERAI | ERAI |
| Northward Near-Surface Wind | vas | $m\ s^{-1}$ | ERAI | ERAI |



The bias correction was performed on the regular 0.5° EWEMBI grid, to which raw CMIP5 GCM data were interpolated with a first-order conservative remapping scheme (Jones, 1999). GCM-to-EWEMBI transfer-function coefficients were calculated based on GCM data from the historical and RCP8.5 CMIP5 experiments representing the periods 1979–2005 and 2006–2013, respectively.

The variables pr, prsn, rlds, sfcWind, tas, tasmax and tasmin were corrected as described by Hempel et al. (2013), except that we defined dry days using a modified threshold value of 0.1 mm/day, since this value was used to correct WFDEI dry-day frequencies (Harris et al., 2013; Weedon et al., 2014). Also, in order to prevent the bias correction from creating unrealistically extreme temperatures, we introduced a maximum value of 3 for the

correction factors of tas – tasmin and tasmax – tas (cf. Hempel et al., 2013, Eq. (25)) and limited tas, tasmin and tasmax to the range [-90°C, 60°C], in line with historical record near-surface temperature observations. Lastly, in order to solve the third of the problems listed above, the methods used to correct rlds and sfcWind in the fast track were equipped with daily (instead of monthly) climatologies using linearly interpolated monthly mean values as in the temperature correction methods (Hempel et al., 2013, Eqs. (16–20)).

Bias-corrected surface pressure was obtained from CMIP5 output of sea level pressure (psl) in three steps. First, EWEMBI ps was reduced to EWEMBI psl using EWEMBI tas, WFDEI and ERAI surface elevation over land except Antarctica and the rest of the earth's surface, respectively, and

$$psl = ps * \exp\left[\frac{g * z}{R * tas}\right], \qquad (1)$$

where z is surface elevation, g is gravity and R is the specific gas constant of dry air. Simulated psl was then corrected using EWEMBI psl and the tas correction method described by Hempel et al. (2013). Finally, the bias-corrected psl was transformed to a bias-corrected ps using (1) with WFDEI and ERAI surface elevation and bias-corrected tas. As alluded to above, hurs and rsds were bias corrected using newly developed methods which respect the lower and upper limits that these variables are exposed to (Lange et al., 2016a). A bias-corrected huss

consistent with bias-corrected hurs, ps and tas was calculated using the equations of Buck (1981) as described in Weedon et al. (2010). The wind components uas and vas were not corrected.

In order to cover the special data needs of FISH-MIP, we additionally provide uncorrected depth-resolved, depth-integrated, surface and bottom oceanic data at monthly temporal resolution for the following variables: Sea

Water X Velocity (uo), Sea Water Y Velocity (vo), Sea Water Temperature (t), Dissolved Oxygen Concentration



(o2), Primary Organic Carbon Production by All Types of Phytoplankton (intpp), Phytoplankton Carbon Concentration (phyc), Small Phytoplankton Carbon Concentration (sphyc), Large Phytoplankton Carbon Concentration (lphyc), Zooplankton Carbon Concentration (zooc), Small Zooplankton Carbon Concentration (szooc), Large Zooplankton Carbon Concentration (lzooc), pH (ph) and Sea Water Salinity (so). Furthermore, the

Tropical Cyclones sector is provided with uncorrected depth-resolved monthly mean Sea Water Potential Temperature (thetao), monthly mean Sea Surface Temperature (tos), monthly mean Air Temperature, and Specific Humidity (ta, hus) at all atmospheric model levels and daily mean Eastward and Northward Wind (ua, va) at 250 and 850 hPa levels.

## 4    Land-use Patterns

The second component of the request for the 1.5°C special report refers to an assessment of "related global greenhouse gas emission pathways". ISIMIP2b will not address this issue by extending the range of potential emission pathways beyond the RCP projections, which provide the basis for the climate model simulations within CMIP5, but rather by assessing the impacts of the socio-economic changes associated with the considered RCPs as far as they are reflected in LU and irrigation changes. To this end, we provide transient LU patterns as

generated by the LU model MAgPIE (Popp et al., 2014a; Stevanović et al., 2016), assuming population growth and economic development as described in SSP2. Such an SSP2 future without explicit mitigation measures for the reduction of greenhouse gas emissions and associated atmospheric concentrations as generated by IAMs is closest to RCP6.0 (Riahi et al., 2016). LU patterns derived by MAgPIE are designed to ensure demand-fulfilling food production where demand is externally prescribed based on an extrapolation of historical relationships

between population and GDP on national levels (Bodirsky et al., 2015). In contrast to the standard SSP scenarios generated within the scenario process (Kriegler et al., 2016), LU changes assessed for ISIMIP2b additionally account for climate and atmospheric $CO_2$ fertilization effects on the underlying patterns of potential crop yields, water availability and terrestrial carbon content. The associated LPJmL crop, water, and biomes simulations are forced by patterns of climate change associated with RCP6.0 and the model also accounts for the $CO_2$ fertilization

effect based on atmospheric concentrations derived from RCP6.0. Potential crop production under rain-fed conditions as well as full irrigation were generated by the global gridded crop component of LPJmL within the ISIMIP fast track (Rosenzweig et al., 2014) and used by MAgPIE to derive LU patterns under cost optimization (see time series of total crop land (irrigated vs. non-irrigated) in Figure 3). Projections of climate change are taken from the four GCMs also used to force the other impacts projections within ISIMIP2b to ensure maximum

consistency. As the MIROC5 climate input data were not part of the ISIMIP fast track, the associated crop yield



projections by LPJmL were added analogously to the fast track simulations to calculate the associated LU patterns.

To reach the low emissions RCP2.6 scenario under SSP2, land-based mitigation measures are of great importance (Popp et al., 2014b). Land-based mitigation in MAgPIE is driven by carbon prices and bioenergy demand from the

REMIND-MAgPIE Integrated Assessment Modelling Framework as implemented in the SSP exercise (Kriegler et al., 2016). REMIND-MAgPIE is designed to derive an optimal mitigation mix under climate-policy settings, maximizing aggregate social consumption across the 21st century. Reaching RCP2.6 from a reference path (i.e. RCP6.0) in an SSP2 setting results in reduced emissions from LU change via avoided deforestation, reduction of non-$CO_2$ emissions from agricultural production, and a strong expansion of bioenergy production combined with carbon

capture and storage (BECCS, see total land area used for second-generation bioenergy production in Figure 3). In this low-emission RCP2.6 scenario (dark blue line in Figure 1 and Figure 3) the underlying patterns of potential crop yields, water availability and terrestrial carbon content are again delivered by LPJmL, but this time forced by RCP2.6 climate change and $CO_2$ concentrations.

The resulting LU and irrigation patterns will be provided as input to the climate-impact models participating in ISIMIP2b, where at least some of the hydrological and global vegetation models are designed to account for these changes (see section 5). In this way, it becomes possible to exemplarily quantify the consequences of the suggested mitigation measures in comparison to the avoided impacts of climate change.

Historical LU changes are used from the HYDE3.2 data (Klein Goldewijk, 2016). For the future projections of LU

patterns, the crop-land and pasture areas in the MAgPIE model are initialized by this historic LU data on the 0.5° grid. Since there is still a transitional difference between historical and projected LU patterns, the period between historic 2010 and projected 2030 patterns is interpolated to achieve a smooth transition. Other land classes in the model (natural and industrial forests and other land pools) are based on Erb et al. (2007) data and harmonized with HYDE3.2 data.

Based on HYDE3.2 we provide patterns for the following categories of agricultural land: 1) irrigated crops, 2) rainfed crops, 3) pasture (intensively managed), and 5) rangeland (less/not managed). From MAgPIE we provide future projection for the following agricultural LU categories: 1) total crop land (rainfed), 2) total crop land (irrigated), 3) bioenergy production (rainfed grass), 4) bioenergy production (rainfed trees), and 5) pasture

(rainfed). As needed by many impact models we also provide a further disaggregation of the agricultural land of both data sets into major food-crop classes based on LUH2 (Land Use Harmonization,





http://luh.umd.edu/data.shtml) and Monfreda et al. (2008) (http://www.earthstat.org/data-download/, see SOM for the technical details).

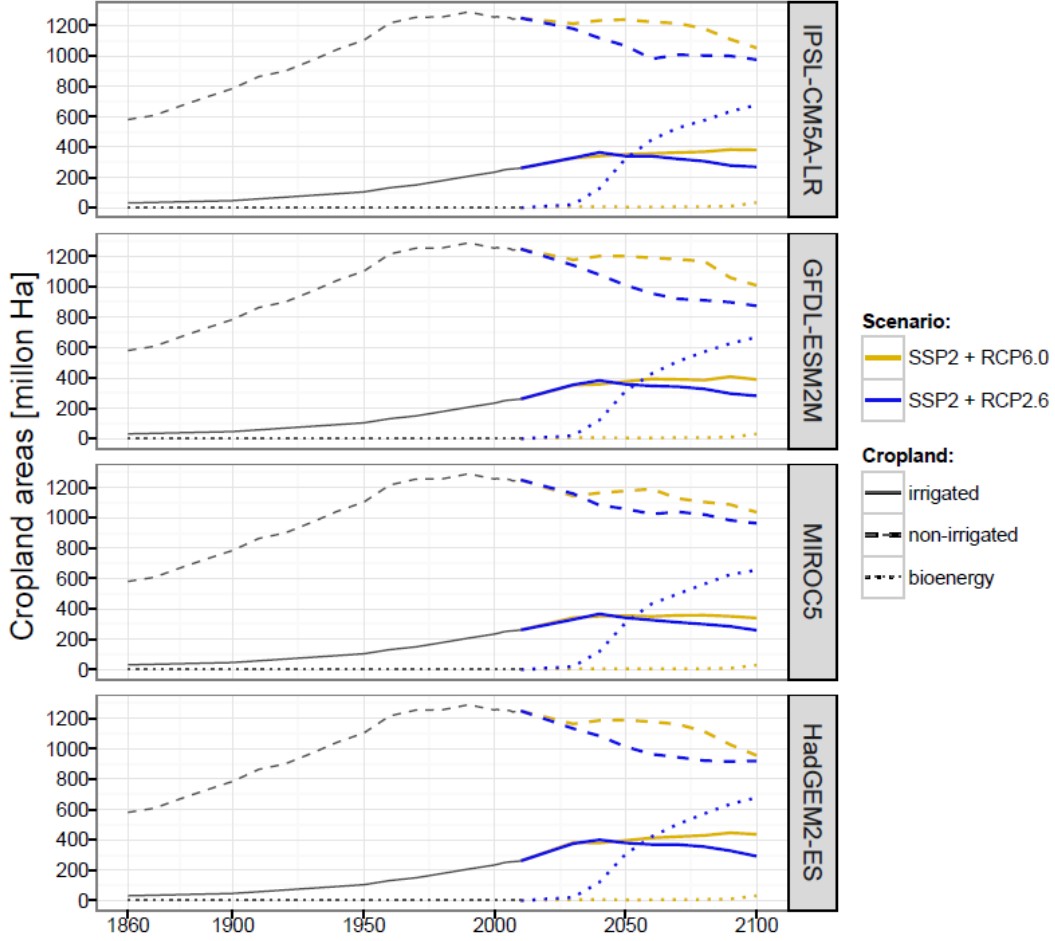

**Figure 3** Time series of total crop land (irrigated (solid lines) and non-irrigated (dashed lines)) as reconstructed for the historical period (1860 - 2010) based on HYDE3.2 (Klein Goldewijk, 2016) and projected under SSP2 (2030-2100) assuming no explicit mitigation of greenhouse gas emissions (RCP6.0, yellow line) and strong mitigation (RCP2.6, dark blue line) as suggested by MAgPIE. Future projections also include land areas for second generation bioenergy production (not included in "total crop land") for the demand generated from the Integrated Assessment Modelling Framework REMIND-MAgPIE, as implemented in the SSP exercise (dotted lines). Global data were linearly interpolated between the historical data set and the projections.

## 5    Patterns of sea-level rise

Sea-level rise is an important factor for climate-change-related impacts on coastal infrastructure. For ISIMIP2b we utilize knowledge on the individual contributions to sea-level rise and construct time series of future total sea-

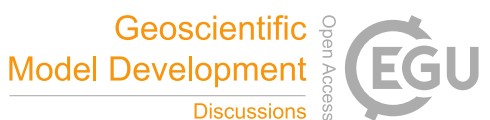

level rise by adding the climate-driven and non-climate-driven contributions. The climate-sensitive components are thermal expansion, mountain glaciers and ice caps, and the large ice sheets on Greenland and Antarctica. We infer thermal expansion directly from the four ISIMIP GCMs. We derive future sea-level rise from mountain glaciers and the Greenland and the Antarctic ice sheets with the "constrained extrapolation" approach (Mengel et al., 2016), driven by the global-mean-temperature evolution of the four ISIMIP GCMs. The approach combines information about long-term sea-level change with observed short-term responses and allows the projection of the different contributions to climate-driven sea-level rise from global-mean-temperature change (see SOM Figure S1 – S5). We add the contribution from glaciers that is not driven by current climate change (Marzeion et al., 2014, see upper panel of Fig. S5 in the SOM) and the contribution from land water storage (Wada et al., 2012, see lower panel of Fig. S5 in the SOM) to yield projections of total sea-level rise. The linear trend of the natural-glacier contribution (Marzeion and Levermann, 2014, Fig. 1c) suggest that the natural contribution reaches zero around year 2056. We therefore approximate this contribution by a parabola with a maximum in 2056, extended with zero trend beyond that year (see SOM, black line in the upper panel of Fig. S5). The land water storage projections are extended to 2299 with the linear 2050-2100 trend.

Past global sea-level rise is available through a meta-analysis of proxy relative sea-level reconstructions (Kopp et al., 2016). We match past observed and future projected total sea level rise by providing both time series relative to the year 2000. We use the observed time series before the year 2000 (Fig. 4, black line) and the projections after that year (Fig. 4, blue (RCP2.6) and yellow (RCP6.0) line).

The spatial pattern of dynamic sea-level changes can be diagnosed directly from the GCMs. We derive the regional variation of sea-level rise from glaciers and the large ice sheets by scaling from their respective gravitational patterns. These patterns are assumed to be time and scenario independent. Total climate-driven sea-level rise at a certain location is the sum of the patterns for dynamic sea-level changes, glaciers and ice caps and the Greenland and the Antarctic ice sheet. For the long-term projections beyond 2100 the constrained extrapolations have been extended to 2299.



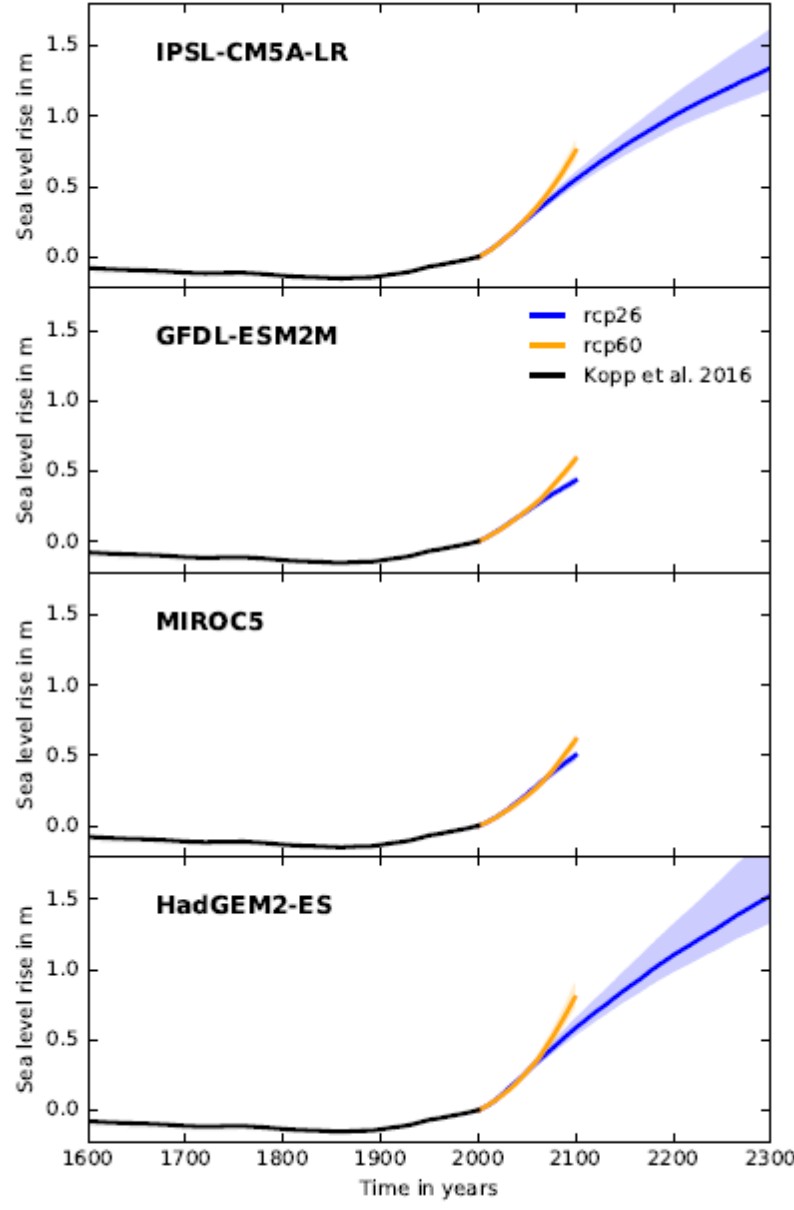

**Figure 4** Time series of global total sea-level rise based on observations (Kopp et al., 2016, black line) until year 2000 and global-mean-temperature change from IPSL-CM5A-LR (panel 1), GFDL-ESM2M (panel 2), MIROC5 (panel 3) and HadGEM2-ES (panel 4) after year 2000: solid lines: Median projections, shaded areas: uncertainty range between the 5[th] and 95[th] percentile
5   of the uncertainty distribution associated with the ice components. Blue: RCP2.6, yellow: RCP6.0. All time series relative to year 2000. Non-climate-driven contribution from glaciers and land water storage are added to the projections.



## 6    Information about population patterns and economic output (Gross Domestic Product, GDP)

We provide population data on a 0.5° grid covering the whole period from 1860 to 2100. The historic data are taken from the HYDE3.2 database (Klein Goldewijk, 2011; Klein Goldewijk et al., 2010). They cover the period 1860 to 2000 in 10-year time steps plus yearly data between 2001 and 2015 with a default resolution of 5'. For the future period, gridded data based on the national SSP2 population projections as described in Samir and Lutz, (2014) are available (Jones and Neill, 2016) covering the period 2010-2100 in 10-year time steps, with a 7.5' resolution. We remap both data sets to the ISIMIP 0.5° grid and interpolate to yearly time steps. In addition, we provide age-specific population data (in 5-year age groups: 0-4, 5-9, etc.) and all-age mortality rates in 5-year time steps on a country level for 2010-2100, corresponding to the same SSP2 projections by Samir and Lutz (2014). Figure 5 shows total global population over time. Both datasets take into account urbanisation trends.

**Figure 5** Time series of global population for the historical period (dots) and future projections following the SSP2 storyline (triangles).

No gridded GDP data are available either for the past or the SSP future projections. For the historical period (1860-2010) we will provide annual country-level data from the Maddison project (Bolt and van Zanden, 2014, www.ggdc.net/maddison/maddison-project/home.htm). Future projections of national GDP will be taken from the SSP database (Dellink et al., 2015, https://secure.iiasa.ac.at/web-apps/ene/SspDb/). The data base includes



country-level GDP projections from 2010-2100 in 10-year time steps. For SSP2, projections from the OECD will be provided.

## 7 Representation of other human influences

There are other human influences that are well documented and partly represented in climate-impact models.
Available indicators of human influences apart from climate change, population changes, changes in national GDP, and LU patterns are primarily: 1) construction of dams and reservoirs, 2) irrigation-water extraction, 3) patterns of inorganic fertilizer application rates, 4) nitrogen deposition, and 5) information about fishing intensities. For all of these input variables, we describe reconstructions to be used for the historical "histsoc" simulations (see Table 2). For models that do not allow for time-varying human influences across the historical period, human influences should be fixed at present-day (year 2005) levels (see dashed line in Figure 1, Group 1) . Beyond 2005 all human influences should be held constant (Group 2) or varied according to SSP2 if associated projections are available (Group 3). Within ISIMIP2b we provide projections of future irrigation-water extraction, fertilizer application rates and nitrogen deposition (see Table 2).

**Table 2** Data sets that will be provided to represent "other human influences" for the historical simulations (histsoc, Group 1) and the future projections accounting for changes in socio-economic drivers (rcp26soc or rcp60soc, Group 2).

| Driver | Historical reconstruction | Future projections |
|---|---|---|
| **Reservoirs & dams** | Includes location, upstream area, capacity, and construction/commissioning year, on a global 0.5° grid.<br>Documentation:<br>http://www.gwsp.org/products/grand-database.html<br><br>**Note:** Simple interpolation can result in inconsistencies between the GranD database and the DDM30 routing network (wrong upstream area due to misaligned dam/reservoir location). We provide a file with locations of all larger dams/reservoirs adapted to DDM30 such as to best match reported upstream areas. | No future data sets are provided. Assumed to be fixed at year 2005 levels. |
| **Water abstraction for domestic & industrial use** | Generated by each modelling group individually (e.g. following the **varsoc** scenario in ISIMIP2a). Modelling groups that do not have their own representation could use an average of the ISIMIP2a | Generated by each modelling group individually.<br><br>Modelling groups that do not |



| | data generated by the other models (available on request). Before 1901 water abstraction for domestic and industrial uses is set to 1901 values. | have their own representation should use the mean (of three models) scenarios for domestic and industrial uses from the Water Futures and Solutions (WFaS; Wada et al., 2016) project consistent with SSP2 and RCP2.6/6.0. |
|---|---|---|
| **Irrigation water extraction (km$^3$)** | Individually derived from the provided land use and irrigation patterns (see section 4)  Water directly used for livestock (e.g. animal husbandry and drinking) except for indirect uses by irrigation of feed crops is expected to be very low (Müller Schmied et al., 2016) and could be set to zero if not directly represented in the individual models. | Derived from future land-use and irrigation patterns provided by MAgPIE (see section 4). Land-use projections are provided for SSP2+RCP6.0 and SSP2+RCP2.6; direct water use for livestock should be ignored (i.e. can be set to zero). |
| **Fertilizer (kg per ha of cropland)** | N fertilizer use (crop specific input per ha of crop land for C$_3$ and C$_4$ annual, C$_3$ and C$_4$ perennial and C$_3$ Nitrogen fixing) is provided for the historical period at an annual time step. This data set is part of the LUH2 dataset developed for CMIP6 (http://luh.umd.edu/data.shtml) based on HYDE3.2 | Inorganic N fertilizer use per area of crop land provided by MAgPIE, different for SSP2+RCP2.6 and SSP2+RCP6.0 |
| **Nitrogen deposition** | Annual, gridded NH$_X$ and NO$_Y$ deposition during 1850-2005 derived from three atmospheric chemistry models (i.e., GISS-E2-R, CCSM-CAM3.5, and GFDL-AM3) in the Atmospheric Chemistry and Climate Model Intercomparison Project (ACCMIP) (0.5° x 0.5°) (Lamarque et al., 2013a, 2013b).  The GISS-E2-R provided monthly nitrogen deposition output; CCSM-CAM3.5 provided monthly nitrogen deposition in each decade from 1850s to the 2000s; and GFDL-AM3 provided monthly nitrogen deposition in five periods (1850-1860, 1871-1950, 1961-1980, 1991-2000, 2001-2010). Annual deposition rates were calculated by aggregating the monthly data, and nitrogen deposition rates in years without model output were calculated according to spline interpolation (CCSM-CAM3.5) or linear interpolation (for GFDL). The original deposition data was downscaled to spatial resolution of half degree (90° N | As per historical reconstruction for 2006-2100 following RCP2.6 and RCP6.0. |



| | to 90° S, 180° W to 180° E) by applying the nearest interpolation. | |
|---|---|---|
| **Fishing intensity** | Depending on model construction, one of: Fishing effort from the Sea Around Us Project (SAUP); catch data from the Regional Fisheries Management Organizations (RFMOs) local fisheries agencies; exponential fishing technology increase and SAUP economic reconstructions.<br><br>Given that the SAUP historical reconstruction starts in 1950, fishing effort should be held at a constant 1950 value from 1860-1950. | held constant after 2005 (**2005soc**) |

## 8  Sector-specific implementation of scenario design

Here we provide a more detailed description of the sector-specific simulations. The grey, red, and blue background colours of the different entries in the tables indicate Group 1, 2, 3 runs, respectively. Runs marked in violet represent additional sector-specific sensitivity experiments. Each simulation run has a name (Experiment I

5   to VII) that is consistent across sectors, i.e. runs from the individual experiments could be combined for a consistent cross-sectoral analysis. Since human influences represented in individual sectors may depend on the RCPs (such as land-use changes), while human influences relevant for other sectors may only depend on the SSP, the number of experiments differs from sector to sector.

### 8.1  Global water

| Climate & CO$_2$ scenarios | |
|---|---|
| **picontrol** | Dynamic pre-industrial climate and 278ppm CO$_2$ concentration |
| **historical** | Historical climate and CO$_2$ concentration. |
| **rcp26** | Future climate and CO$_2$ concentration from RCP2.6 |
| **rcp60** | Future climate and CO$_2$ concentration from RCP6.0 |
| Human influence and land-use scenarios | |
| **1860soc** | Pre-industrial land use and other human influences. Given the small effect of dams & reservoirs before 1900, modellers may apply the 1901 dam/reservoir configuration during the pre-industrial period and the 1861-1900 part of the historical period if that is significantly easier than applying the 1861 configuration. |



| | |
|---|---|
| **histsoc** | Varying historical land use and other human influences. |
| **2005soc** | Fixed year-2005 land use and other human influences. |
| **rcp26soc** | Varying land use, water abstraction and other human influences according to SSP2 and RCP2.6; fixed year-2005 dams and reservoirs. For models using fixed LU types, varying irrigation areas can also be considered as varying land use. |
| **rcp60soc** | Varying land use, water abstraction and other human influences according to SSP2 and RCP6.0, fixed year-2005 dams and reservoirs. For models using fixed LU types, varying irrigation areas can also be considered as varying land use. |
| **2100rcp26soc** | Land use and other human influences fixed at year 2100 levels according to RCP2.6. |

**Table 3** ISIMIP2b scenario specification for the global water model simulations. Option 2* only if option 1 not possible.

| | Experiment | Input | pre-industrial 1661-1860 | historical 1861-2005 | future 2006-2100 | extended future 2101-2299 |
|---|---|---|---|---|---|---|
| **I** | no climate change, pre-industrial $CO_2$ | Climate & $CO_2$ | **picontrol** | **picontrol** | **picontrol** | **picontrol** |
| | varying LU & human influences up to 2005, then fixed at 2005 levels thereafter | Human & LU | Option 1: **1860soc** / Option 2*: **2005soc** | Option 1: **histsoc** / Option 2*: **2005soc** | **2005soc** | **2005soc** |
| **II** | RCP2.6 climate & $CO_2$ | Climate & $CO_2$ | Experiment I | **historical** | **rcp26** | **rcp26** |
| | varying LU & human influences up to 2005, then fixed at 2005 levels thereafter | Human & LU | | Option 1: **histsoc** / Option 2*: **2005soc** | **2005soc** | **2005soc** |
| **III** | RCP6.0 climate & $CO_2$ | Climate & $CO_2$ | Experiment I | Experiment II | **rcp60** | not simulated |
| | varying LU & human influences up to 2005, then fixed at 2005 levels thereafter | Human & LU | | | **2005soc** | |
| **IV** | no climate change, pre-industrial $CO_2$ | Climate & $CO_2$ | Experiment I | Experiment I | **picontrol** | **picontrol** |



| | | | | | | |
|---|---|---|---|---|---|---|
| | varying human influences & LU up to 2100 (RCP2.6), then fixed at 2100 levels thereafter | Human & LU | | | **rcp26soc** | **2100rcp26soc** |
| **V** | no climate change, pre-industrial CO$_2$ | Climate & CO$_2$ | Experiment I | Experiment I | **picontrol** | not simulated |
| | varying human influences & LU (RCP6.0) | Human & LU | | | **rcp60soc** | |
| **VI** | RCP2.6 climate & CO$_2$ | Climate & CO$_2$ | Experiment I | Experiment II | **rcp26** | **rcp26** |
| | varying human influences & LU up to 2100 (RCP2.6), then fixed at 2100 levels thereafter | Human & LU | | | **rcp26soc** | **2100rcp26soc** |
| **VII** | RCP6.0 climate & CO$_2$ | Climate & CO$_2$ | Experiment I | Experiment II | **rcp60** | not simulated |
| | varying human influences & LU (RCP6.0) | Human & LU | | | **rcp60soc** | |

For the historical period, groups that have limited computational capacities may choose to report only part of the full period, but including at least 1961-2005. All other periods should be reported completely. For those models that do not represent *changes* in human impacts, those impacts should be held fixed at 2005 levels throughout all
Group 1 (cf. "2005soc" marked as dashed blue lines in Figure 1) and Group 2 simulations. Group 3 will be identical to Group 2 for these models and thus does not require additional simulations. Models that do not include human impacts *at all* are asked to run the Group 1 and Group 2 simulations nonetheless, since these simulations will still allow for an exploration of the effects of climate change compare to pre-industrial climate, and will also allow for a better assessment of the relative importance of human impacts versus climate impacts. These runs should be
named as "nosoc" simulations.

**8.2    Regional water**

The regional-scale simulations are performed for 12 large river basins. In six river basins (Tagus, Niger, Blue Nile, Ganges, Upper Yangtze and Darling) water management (dams/reservoirs, water abstraction) will be implemented. In the other six river basins, human influences such as LU changes, dams and reservoirs, and water
abstraction is not relevant (MacKenzie, Upper Yellow, Upper Amazon) or negligible (Rhine, Lena, Upper



Mississippi), and can be ignored. Apart from this, regional water simulations should follow the global water simulations to allow for a cross-scale comparison of the simulations.

## 8.3    Biomes

Since the pre-industrial simulations are an important part of the experiments, the spin-up has to finish before the
5    pre-industrial simulations start. The spin-up should be using pre-industrial climate (picontrol) and year 1860 levels of "other human influences". For this reason, the pre-industrial climate data should be replicated as often as required.

| Climate & $CO_2$ scenarios | |
|---|---|
| **picontrol** | Pre-industrial climate and 278ppm $CO_2$ concentration |
| **historical** | Historical climate and $CO_2$ concentration. |
| **rcp26** | Future climate and $CO_2$ concentration from RCP2.6 |
| **rcp60** | Future climate and $CO_2$ concentration from RCP6.0 |
| **2005co2** | CO2 concentration fixed at 2005 levels at 378.81ppm.. |
| **Human influence  and land-use scenarios** | |
| **1860soc** | Constant pre-industrial (1860) land use, nitrogen deposition, and fertilizer input. |
| **histsoc** | Varying historical land use, nitrogen deposition and fertilizer input. |
| **2005soc** | Fixed year-2005 land use, nitrogen deposition and fertilizer input. |
| **rcp26soc** | Varying land use, water abstraction, nitrogen deposition and fertilizer input according to SSP2 and RCP2.6. |
| **rcp60soc** | Varying land use, water abstraction, nitrogen deposition and fertilizer input according to SSP2 and RCP6.0. |
| **2100rcp26soc** | Land use, nitrogen deposition and fertilizer input fixed at year 2100 levels according to RCP2.6 in 2100. |

**Table 4:** ISIMIP2b scenario specification for the global biomes model simulations.

| | Experiment | Input | Pre-industrial 1661-1860 | Historical 1861-2005 | Future 2006-2100 | Extended future 2101-2299 |
|---|---|---|---|---|---|---|
| **I** | no climate change, pre-industrial $CO_2$ | Climate & $CO_2$ | **picontrol** | **picontrol** | **picontrol** | **picontrol** |
| | varying LU & human influences up to 2005, then fixed at 2005 levels | Human & LU | **1860soc** | **histsoc** | **2005soc** | **2005soc** |





| | | | | | | |
|---|---|---|---|---|---|---|
| | thereafter | | | | | |
| **II** | RCP2.6 climate & $CO_2$ | Climate & $CO_2$ | Experiment I | | **historical** | **rcp26** | **rcp26** |
| | varying LU & human influences up to 2005, then fixed at 2005 levels thereafter | Human & LU | | | **histsoc** | **2005soc** | **2005soc** |
| **IIa** | RCP2.6 climate, $CO_2$ after 2005 fixed at 2005 levels | Climate & $CO_2$ | Experiment I | Experiment II | **rcp26, 2005co2** | **rcp26, 2005co2** |
| | varying LU & human influences up to 2005, then fixed at 2005 levels thereafter | Human & LU | | | **2005soc** | **2005soc** |
| **III** | RCP6.0 climate & $CO_2$ | Climate & $CO_2$ | Experiment I | Experiment II | **rcp60** | not simulated |
| | varying LU & human influences up to 2005, then fixed at 2005 levels thereafter | Human & LU | | | **2005soc** | |
| **IV** | no climate change, pre-industrial $CO_2$ | Climate & $CO_2$ | Experiment I | Experiment I | **picontrol** | **picontrol** |
| | varying human influences & LU up to 2100 (RCP2.6), then fixed at 2100 levels thereafter | Human & LU | | | **rcp26soc** | **2100rcp26soc** |
| **V** | no climate change, pre-industrial $CO_2$ | Climate & $CO_2$ | Experiment I | Experiment I | **picontrol** | not simulated |
| | varying human influences & LU (RCP6.0) | Human & LU | | | **rcp60soc** | |
| **VI** | RCP2.6 climate & $CO_2$ | Climate & $CO_2$ | Experiment I | Experiment II | **rcp26** | **rcp26** |
| | varying human influences & LU up to 2100 (RCP2.6), then fixed at 2100 levels thereafter | Human & LU | | | **rcp26soc** | **2100rcp26soc** |
| **VII** | RCP6.0 climate & $CO_2$ | Climate & $CO_2$ | Experiment I | Experiment II | **rcp60** | not simulated |



| varying human influences & LU (RCP6.0) | Human & LU | | | **rcp60soc** | |
|---|---|---|---|---|---|

## 8.4    Regional Forestry

Entries marked in light grey specify the *starting* conditions on which the runs should be based, since several experiments branch off after the Group 1 runs (meaning that these runs do not have to be repeated).

| **Climate scenarios** | |
|---|---|
| **picontrol** | Pre-industrial climate and 278ppm $CO_2$ concentration. |
| **historical** | Historical climate and $CO_2$ concentration. |
| **rcp26** | Future climate and $CO_2$ concentration from RCP2.6. |
| **rcp60** | Future climate and $CO_2$ concentration from RCP6.0. |
| **2005co2** | CO2 concentration fixed at 2005 levels at 378.81ppm. |
| **Human influences scenarios** | |
| **histsoc** | Manage future forests according to historical management guidelines without species change and keeping the same rotation length and thinning types. |
| **2005soc** | **Business as usual (BAU)**: Manage future forests according to present-day management guidelines without species change and keeping the same rotation length and thinning types. |
| **AMsoc** | **Assisted migration**: Manage future forests according to BAU until final harvest but then plant tree species that would be the natural vegetation under the projected climate change according to species-distribution model predictions (Hanewinkel et al., 2012). |
| **BEsoc** | **Bioenergy**: Manage future forests by changing the main tree species to a species that is very productive with shorter rotations and thinning from below to increase the production of wood for bioenergy. |
| **BAUAsoc** | **BAU_adapted**: manage future forests by changing the species to a species that produces similar services as the BAU forests but with adjusted rotations lengths, species change and thinning from above. |



**Table 4** ISIMIP2b scenarios for the regional forest simulations.

| | Experiment | Input | Pre-industrial 1661-1860 | Historical 1861-2005 | Future 2006-2100 | Extended future 2101-2299 |
|---|---|---|---|---|---|---|
| **I** | no climate change, pre-industrial $CO_2$ | Climate & $CO_2$ | not simulated | | **picontrol** | **picontrol** | **picontrol** |
| | present-day management (BAU) | Human & LU | | | **histsoc** | **2005soc** | **2005soc** |
| **II** | RCP2.6 climate & $CO_2$ | Climate & $CO_2$ | not simulated | | **historical** | **rcp26** | **rcp26** |
| | present-day management (BAU) | Human & LU | | | **histsoc** | **2005soc** | **2005soc** |
| **IIa** | RCP2.6 climate, $CO_2$ fixed after 2005 | Climate & $CO_2$ | not simulated | Experiment II | **rcp26, 2005co2** | **rcp26, 2005co2** |
| | present-day management (BAU) | Human & LU | | | **2005soc** | **2005soc** |
| **III** | RCP6.0 climate & $CO_2$ | Climate & $CO_2$ | not simulated | Experiment II | **rcp60** | not simulated |
| | present-day management (BAU) | Human & LU | | | **2005soc** | |
| **IIIa** | RCP6.0 climate, $CO_2$ fixed after 2005 | Climate & $CO_2$ | not simulated | Experiment II | **rcp60, 2005co2** | not simulated |
| | present-day management (BAU) | Human & LU | | | **2005soc** | |
| **IVa** | no climate change, pre-industrial $CO_2$ | Climate & $CO_2$ | not simulated | Experiment I | **picontrol** | **picontrol** |
| | varying management (AM) | Human & LU | | | **AMsoc** | **AMsoc** |
| **IVb** | no climate change, pre-industrial $CO_2$ | Climate & $CO_2$ | not simulated | Experiment I | **picontrol** | **picontrol** |
| | varying management (BE) | Human & LU | | | **BEsoc** | **BEsoc** |
| **IVc** | no climate change, pre-industrial $CO_2$ | Climate & $CO_2$ | not simulated | Experiment I | **picontrol** | **picontrol** |





| | | | | | | |
|---|---|---|---|---|---|---|
| | varying management (BAUA) | Human & LU | | | **BAUAsoc** | **BAUAsoc** |
| **VIa** | RCP2.6 climate & $CO_2$ | Climate & $CO_2$ | not simulated | Experiment II | **rcp26** | **rcp26** |
| | varying management (AM) | Human & LU | | | **AMsoc** | **AMsoc** |
| **VIb** | RCP2.6 climate & $CO_2$ | Climate & $CO_2$ | not simulated | Experiment II | **rcp26** | **rcp26** |
| | varying management (BE) | Human & LU | | | **BEsoc** | **BEsoc** |
| **VIc** | RCP2.6 climate & $CO_2$ | Climate & $CO_2$ | not simulated | Experiment II | **rcp26** | **rcp26** |
| | varying management (BAUA) | Human & LU | | | **BAUAsoc** | **BAUAsoc** |
| **VIIa** | RCP6.0 climate & $CO_2$ | Climate & $CO_2$ | not simulated | Experiment II | **rcp60** | not simulated |
| | varying management (AM) | Human & LU | | | **AMsoc** | |
| **VIIb** | RCP6.0 climate & $CO_2$ | Climate & $CO_2$ | not simulated | Experiment II | **rcp60** | not simulated |
| | varying management (BE) | Human & LU | | | **BEsoc** | |
| **VIIc** | RCP6.0 climate & $CO_2$ | Climate & $CO_2$ | not simulated | Experiment II | **rcp60** | not simulated |
| | varying management (BAUA) | Human & LU | | | **BAUAsoc** | |

### 8.5   Permafrost

Permafrost modules that are embedded into global biomes models should report the permafrost variables for the biomes model simulations specified in section 7.3 + the extension beyond 2299 described in table 6. Table 6

5   describes the simulations for models only participating as permafrost models, assuming that for the relevant regions "other human influences" only play a minor role, i.e. the regional simulations can be done as "naturalized" runs (nosoc).

| Climate & $CO_2$ scenarios | |
|---|---|
| **picontrol** | Pre-industrial climate and 278ppm $CO_2$ concentration |
| **historical** | Historical climate and $CO_2$ concentration. |



| rcp26 | Future climate and $CO_2$ concentration from RCP2.6 |
|---|---|
| rcp60 | Future climate and $CO_2$ concentration from RCP6.0 |
| 2299rcp26 | Repeating climate between 2270 and 2299 for additional 200 years up to 2500 (or equilibrium if possible), $CO_2$ fixed at year 2299 levels |
| 2005co2 | Fixed year 2005 $CO_2$ concentration at 378.81ppm. |
| **Human influence & land-use scenarios** | |
| nosoc | No human influences |

**Table 5** ISIMIP2b scenario specification for the permafrost simulations.

| | Experiment | Input | Pre-industrial 1661-1860 | Historical 1861-2005 | Future 2006-2100 | Extended future 2101-2299 | Beyond 2299 |
|---|---|---|---|---|---|---|---|
| **I** | no climate change, pre-industrial $CO_2$ | Climate & $CO_2$ | **picontrol** | not simulated | not simulated | not simulated | not simulated |
| | no other human influences | Human & LU | **nosoc** | | | | |
| **II** | RCP2.6 climate & $CO_2$ | Climate & $CO_2$ | Experiment I | **historical** | **rcp26** | **rcp26** | **2299rcp26** |
| | no other human influences | Human & LU | | **nosoc** | **nosoc** | **nosoc** | **nosoc** |
| **IIa** | RCP6.0 climate, $CO_2$ varying until 2005, then fixed at 2005 levels thereafter | Climate & $CO_2$ | Experiment I | Experiment II | **rcp26, 2005co2** | **rcp26, 2005co2** | **2299rcp26, 2005co2** |
| | no other human influences | Human & LU | | | **nosoc** | **nosoc** | **nosoc** |
| **III** | RCP2.6 climate & $CO_2$ | Climate & $CO_2$ | Experiment I | Experiment II | **rcp60** | not simulated | not simulated |
| | no other human influences | Human & LU | | | **nosoc** | | |

## 8.6 Agriculture

5  Crop-model simulations should be provided as pure crop runs (i.e. assuming that each crop grows everywhere). In this future LU pattern can be applied in post-processing ensuring maximum flexibility. Simulations should be



provided for the four major crops (wheat, maize, soy, and rice). For each crop there should be a full irrigation run (firr) and a no-irrigation run (noirr), as specified for the ISIMIP fast track.

Those models that cannot simulate time varying management/human impacts/fertilizer input should keep these fixed at year 2005 levels throughout the simulations ("2005soc" scenario in Group 1 (dashed line in Figure 1 a)

5 and "2005soc" scenario in Group 2). They only need to run the first preindustrial period of Experiment I (1661-1860). Group 3 runs only refer to models that are able to represent future changes in human management (varying crop varieties or fertilizer input).

| Climate & $CO_2$ scenarios | |
|---|---|
| **picontrol** | Pre-industrial climate and 278ppm $CO_2$ concentration |
| **historical** | Historical climate and $CO_2$ concentration. |
| **rcp26** | Future climate and $CO_2$ concentration from RCP2.6 |
| **rcp60** | Future climate and $CO_2$ concentration from RCP6.0 |
| **Human influence & land-use scenarios** | |
| **1860soc** | Pre-industrial levels of fertilizer input. |
| **histsoc** | Varying historical fertilizer input. |
| **2005soc** | Fixed year 2005 management |
| **2005co2** | Fixed year 2005 levels of $CO_2$ at 378.81ppm. |
| **rcp26soc** | Varying level of fertilizer input and varying crop varieties associated with SSP2 and RCP2.6 |
| **rcp60soc** | Varying level of fertilizer input and varying crop varieties associated with SSP2 and RCP6.0 |
| **2100rcp26soc** | Fertilizer input and crop varieties fixed at year 2100. |

**Table 6** ISIMIP2b scenario specification for the global crop model simulations. Option 2* only if option 1 not possible.

| | Experiment | Input | Pre-industrial 1661-1860 | Historical 1861-2005 | Future 2006-2100 | Extended future 2101-2299 |
|---|---|---|---|---|---|---|
| **I** | no climate change, pre-industrial $CO_2$ | Climate & $CO_2$ | **picontrol** | **picontrol** | **picontrol** | **picontrol** |
| | varying management until 2005, then fixed at 2005 levels thereafter | Human & LU | Option 1*: **1860soc** | Option 1*: **histsoc** | **2005soc** | **2005soc** |
| | | | Option 2*: **2005soc** | Option 2*: **2005soc** | | |
| **II** | RCP2.6 climate & $CO_2$ | Climate & $CO_2$ | Experiment I | **historical** | **rcp26** | **rcp26** |





| | | | | | | |
|---|---|---|---|---|---|---|
| | varying management until 2005, then fixed at 2005 levels thereafter | Human & LU | | Option 1*: **histsoc** / Option 2*: **2005soc** | **2005soc** | **2005soc** |
| **IIa** | RCP2.6 climate, CO$_2$ after 2005 fixed at 2005 levels | Climate | Experiment I | Experiment II | **rcp26, 2005co2** | **rcp26, 2005co2** |
| | varying management until 2005, then fixed at 2005 levels thereafter | Human & LU | | | **2005soc** | **2005soc** |
| **III** | RCP6.0 climate & CO$_2$ | Climate & CO$_2$ | Experiment I | Experiment II | **rcp60** | not simulated |
| | varying management until 2005, then fixed at 2005 levels thereafter | Human & LU | | | **2005soc** | |
| **IV** | no climate change, pre-industrial CO$_2$ | Climate & CO$_2$ | Experiment I | Experiment I | **picontrol** | **picontrol** |
| | varying management up to 2100 (RCP2.6), then fixed at 2100 levels thereafter | Human & LU | | | **rcp26soc** | **2100rcp26soc** |
| **V** | no climate change, pre-industrial CO$_2$ | Climate & CO$_2$ | Experiment I | Experiment II | **picontrol** | not simulated |
| | varying management (RCP6.0) | Human & LU | | | **rcp60soc** | |
| **VI** | RCP2.6 climate & CO$_2$ | Climate & CO$_2$ | Experiment I | Experiment II | **rcp26** | **rcp26** |
| | varying management up to 2100 (RCP2.6), then fixed at 2100 levels thereafter | Human & LU | | | **rcp26soc** | **2100rcp26soc** |
| **VII** | RCP6.0 climate & CO$_2$ | Climate & CO$_2$ | Experiment I | Experiment II | **rcp60** | |
| | varying management (RCP6.0) | Human & LU | | | **rcp26soc** | |



## 8.7    Energy

Those models that do not account for varying societal conditions (population, GDP, etc.) should keep these fixed at year 2005 levels throughout the simulations ("2005soc" scenario in Group 1 and Group 2). However, renewable energy capacities have grown strongly between 2005 and today and the fixing to 2005 capacities refers to a bad

5    representation of present day capacities, therefore making it difficult to come to sound conclusions about the influence of climate change. Therefore an additional scenario is added, fixing the installed renewable power generation and other socio-economic drivers to the year 2015. These scenarios should be named "2015soc". Those models that do not account for varying societal conditions only need to run the first pre-industrial period of Experiment I (1661-1860, see option 2 of Experiment I below). The models focusing on the simulation of future

10    projections (e.g. some IAMs) need to run experiment variations associated only with the periods post-2006. Group 3 runs are only relevant for models that are able to represent future changes in societal conditions.

| Climate & CO$_2$ scenarios | |
|---|---|
| **picontrol** | Dynamic pre-industrial climate |
| **historical** | Historical climate and CO$_2$ concentration. |
| **rcp26** | Future climate and CO$_2$ concentration from RCP2.6 |
| **rcp60** | Future climate and CO$_2$ concentration from RCP6.0 |
| **Human influence & land-use scenarios** | |
| **1860soc** | Pre-industrial society |
| **histsoc** | Varying society |
| **2005soc** | Representation of fixed year 2005 society |
| **2015soc** | Representation of fixed year 2015 society |
| **rcp26soc** | Varying society according to SSP2+RCP2.6 |
| **rcp60soc** | Varying society according to SSP2+RCP6.0 |
| **2100rcp26soc** | Representation of fixed year 2100 society according to the last year of rcp26soc. |

**Table 7** ISIMIP2b scenarios energy sector simulations. Option 2* only if option 1 not possible.

| | Experiment | Input | Pre-industrial 1661-1860 | Historical 1861-2005 | Future 2006-2100 | Extended future 2101-2299 |
|---|---|---|---|---|---|---|
| **I** | no climate change, pre-industrial CO$_2$ | Climate & CO$_2$ | **picontrol** | **picontrol** | **picontrol** | **picontrol** |
| | varying society up to 2005, then fixed at 2005 levels thereafter | Human & LU | Option 1: **1860soc** | Option 1: **histsoc** | **2005soc** | **2005soc** |





| | | | Option 2*: 2005soc | Option 2*: 2005soc | | |
|---|---|---|---|---|---|---|
| **Ia** | no climate change, pre-industrial CO$_2$ | Climate & CO$_2$ | **picontrol** | **picontrol** | **picontrol** | **picontrol** |
| | varying society up to 2015, then fixed at 2015 levels thereafter | Human & LU | Option 1: **1860soc**<br>Option 2*: **2015soc** | Option 1: **histsoc**<br>Option 2*: **2015soc** | **2015soc** | **2015soc** |
| **II** | RCP2.6 climate & CO$_2$ | Climate & CO$_2$ | Experiment I | **historical** | **rcp26** | **rcp26** |
| | varying society up to 2005, then fixed at 2005 levels thereafter | LU etc. | | Option 1: **histsoc**<br>Option 2*: **2005soc** | **2005soc** | **2005soc** |
| **IIa** | RCP2.6 climate & CO$_2$ | Climate & CO$_2$ | Experiment Ia | **historical** | **rcp26** | **rcp26** |
| | varying society up to 2015, then fixed at 2015 levels thereafter | Human & LU | | Option 1: **histsoc**<br>Option 2*: **2015soc** | **2015soc** | **2015soc** |
| **III** | RCP6.0 climate & CO$_2$ | Climate & CO$_2$ | Experiment I | Experiment II | **rcp60** | not simulated |
| | varying society up to 2005, then fixed at 2005 levels thereafter | LU etc. | | | **2005soc** | |
| **IIIa** | RCP6.0 climate & CO$_2$ | Climate & CO$_2$ | Experiment Ia | Experiment IIa | **Rcp60** | not simulated |
| | varying society up to 2015, then fixed at 2015 levels thereafter | Human & LU | | | **2015soc** | |
| **IV** | no climate change, pre-industrial CO$_2$ | Climate & CO$_2$ | Experiment I | Experiment I | **picontrol** | **picontrol** |
| | varying society up to 2100 (SSP2+RCP2.6), then fixed at 2100 levels thereafter | LU etc. | | | **rcp26soc** | **2100rcp26soc** |




| | | | | | | |
|---|---|---|---|---|---|---|
| **V** | no climate change, pre-industrial $CO_2$ | Climate | Experiment I | Experiment II | **picontrol** | not simulated |
| | varying society up to 2100 (SSP2+RCP6.0), then fixed at 2100 levels thereafter | LU etc. | | | **rcp60soc** | |
| **VI** | RCP6.0 climate & $CO_2$ | Climate | Experiment I | Experiment II | **rcp26** | **rcp26** |
| | varying society up to 2100 (SSP2+RCP2.6), then fixed at 2100 levels thereafter | LU etc. | | | **rcp26soc** | **2100rcp26soc** |
| **VII** | RCP6.0 climate & $CO_2$ | Climate | Experiment I | Experiment II | **rcp60** | |
| | varying society up (SSP2+RCP6.0) | LU etc. | | | **rcp26soc** | |

## 8.8    Health

The following protocol has been designed for contributions on temperature-related mortality (TRM). Other climate-sensitive health outcomes (e.g. vector-borne diseases) have not been considered yet. However,
5    contributions focusing on other health outcomes are of course highly welcome as long as the general setup of the cross-sectoral ISIMIP2b protocol is followed. Those TRM models that do not account for varying societal conditions (population, GDP, mortality baselines, etc.) should keep these fixed at year 2005 levels throughout the simulations ("2005soc" scenario in Group 1 (dashed line in Figure 1 a) and Group 2). They only need to run the first preindustrial period of Experiment I (1661-1860). Group 3 runs only refer to models that are able to
10    represent future changes in societal conditions.

| Climate & $CO_2$ scenarios | |
|---|---|
| **picontrol** | Dynamic pre-industrial climate |
| **historical** | Historical climate and $CO_2$ concentration. |
| **rcp26** | Future climate and $CO_2$ concentration from RCP2.6 |
| **rcp60** | Future climate and $CO_2$ concentration from RCP6.0 |
| Human influence & land-use scenarios | |
| **1860soc** | Pre-industrial society |
| **2005soc** | Representation of fixed year 2005 society |
| **ssp2soc** | Varying society according to SSP2 |





| 2100ssp2soc | Representation of fixed year 2100 society according to SSP2 |
|---|---|
| **no adaptation** | present-day exposure-response relationship |
| **with adaptation** | altered exposure-response relationships (shifted minimum mortality temperature/slopes) |
| **('_adapt')** | according to default adaptation assumptions. |

**Table 8** ISIMIP2b scenario specification for the simulations of temperature-related mortality.

| | Experiment | Input | Pre-industrial 1661-1860 | Historical 1861-2005 | Future 2006-2100 | Extended future 2101-2299 |
|---|---|---|---|---|---|---|
| **I** | no climate change, pre-industrial $CO_2$ | Climate & $CO_2$ | **picontrol** | **picontrol** | **picontrol** | **picontrol** |
| | varying society up to 2005, then fixed at 2005 levels thereafter, no adaptation | Human & LU | Option 1:**1860soc** / Option 2*: **2005soc** | Option 1: **histsoc** / Option 2*: **2005soc** | **2005soc** | **2005soc** |
| **II** | RCP2.6 climate & $CO_2$ | Climate & $CO_2$ | Experiment I | **historical** | **rcp26** | **rcp26** |
| | varying society up to 2005, then fixed at 2005 levels thereafter, no adaptation | Human & LU | Experiment I | Option 1*: **histsoc** / Option 2*: **2005soc** | **2005soc** | **2005soc** |
| **IIa** | RCP2.6 climate & $CO_2$ | Climate & $CO_2$ | Experiment I | Experiment II | **rcp26** | **rcp26** |
| | varying society up to 2005, then fixed at 2005 levels thereafter, with adaptation | Human & LU | Experiment I | Experiment II | **2005soc_adapt** | **2005soc_adapt** |
| **III** | RCP6.0 climate & $CO_2$ | Climate & $CO_2$ | Experiment I | Experiment II | **rcp60** | not simulated |
| | varying society up to 2005, then fixed at 2005 levels thereafter, no adaptation | Human & LU | Experiment I | Experiment II | **2005soc** | not simulated |



| | | | | | | |
|---|---|---|---|---|---|---|
| **IIIa** | RCP6.0 climate & $CO_2$ | Climate & $CO_2$ | Experiment I | Experiment II | **rcp60** | not simulated |
| | varying society up to 2005, then fixed at 2005 levels thereafter, with adaptation | Human & LU | | | **2005soc_adapt** | |
| **IV** | no climate change, pre-industrial $CO_2$ | Climate & $CO_2$ | Experiment I | Experiment I | **picontrol** | **picontrol** |
| | varying society (SSP2) up to 2100, then fixed at 2100 levels thereafter, no adaptation | Human & LU | | | **rcp26soc** | **2100rcp26soc** |
| **VI** | RCP2.6 climate & $CO_2$ | Climate & $CO_2$ | Experiment I | Experiment II | **rcp26** | **rcp26** |
| | varying society (SSP2) up to 2100, then fixed at 2100 levels thereafter, no adaptation | Human & LU | | | **rcp26soc** | **2100rcp26soc** |
| **VIa** | RCP2.6 climate & $CO_2$ | Climate & $CO_2$ | Experiment I | Experiment II | **rcp26** | **rcp26** |
| | varying society (SSP2) up to 2100, with adpatation | Human & LU | | | **ssp2soc_adapt** | **2100ssp2soc** |
| **VII** | RCP6.0 climate & $CO_2$ | Climate & $CO_2$ | Experiment I | Experiment II | **rcp60** | not simulated |
| | varying society (SSP2), no adaptation | Human & LU | | | **ssp2soc** | |
| **VII a** | RCP6.0 climate & $CO_2$ | Climate & $CO_2$ | Experiment I | Experiment II | **rcp60** | not simulated |
| | varying society (SSP2), with adaptation | Human & LU | | | **ssp2soc_adapt** | |





### 8.9    Coastal Infrastructure

Climate change affects coastal infrastructure through rising mean and extreme sea levels, causing damages through temporary flooding and losses due to permanent submergence of land. To assess these impacts, climate scenarios have to be complemented by sea-level-rise projections. While the information about thermal expansion and dynamical changes of sea level is provided by the four GCMs considered, contributions from mountain glaciers and ice sheets have to be added from other sources, which introduces a further dimension of uncertainty (see section 5). The uncertainty range introduced is substantial and a least on equal footing with the climate model and scenario uncertainty (e.g. Kopp et al. 2014). To reflect this aspect we include an additional scenario dimension in the scenario design for this sector and sample this by providing projections for the median and $5^{th}$ and $95^{th}$ percentiles of the contributions from ice sheets and mountain glaciers to sea-level rise. One aspect specific to the coastal-infrastructure sector is that impacts are extremely non-linear in and sensitive to adaptation. Impacts without adaptation are 2-3 orders of magnitudes higher than those with adaptation (Hinkel et al. 2014). This leads to the circumstance that the regions with the highest infrastructure damages under the scenarios without adaptation are actually the regions least vulnerable to sea-level rise, because it is highly cost-efficient and standard practise to protect those regions against sea-level rise. Scenarios including adaptation are therefore added to the protocol to provide projections of climate change risks including adaptation potentials.

Those models that do not account for varying societal conditions (population, GDP, protection levels etc.) should keep these fixed at year 2005 levels throughout the simulations ("2005soc" scenario in Group 1 (dashed line in Figure 1a) + "rcp26soc"/"rcp60soc" scenario in Group 2). They only need to run the first pre-industrial period of Experiment I (1661-1860). Group 3 runs are only for models able to represent future societal changes.

| Climate & CO$_2$ scenarios | |
|---|---|
| **picontrol** | Pre-industrial climate |
| **historical** | Historical climate and CO$_2$ concentration. |
| **rcp26** | Future climate and CO$_2$ concentration from RCP2.6 |
| **rcp60** | Future climate and CO$_2$ concentration from RCP6.0 |
| **Human influence & land-use scenarios** | |
| **1860soc** | Pre-industrial society and protection |
| **2005soc** | Representation of fixed year 2005 society and protection |
| **ssp2soc** | Varying society and protection according to SSP2 |
| **2100ssp2soc** | Representation of fixed year 2100 society and protection according to SSP2 |





Table 9 ISIMIP2b scenario specification for the simulations of impacts on coastal infrastructure. Option 2* only if option 1 not possible.

| | Experiment | Input | Pre-industrial 1661-1860 | Historical 1861-2005 | Future 2006-2100 | Extended future 2101-2299 |
|---|---|---|---|---|---|---|
| **I** | no climate change, pre-industrial CO$_2$ | Climate & CO$_2$ | picontrol | picontrol | picontrol | picontrol |
| | varying society & protection up to 2005, then fixed at 2005 levels thereafter | Human & LU | Option 1:**1860soc** / Option 2*: **2005soc** | Option 1: **histsoc** / Option 2*: **2005soc** | 2005soc | 2005soc |
| **II** | RCP2.6 climate & CO$_2$ | Climate & CO$_2$ | Experiment I | historical | rcp26 | rcp26 |
| | varying society & protection up to 2005, then fixed at 2005 levels thereafter | Human & LU | | Option 1*: **histsoc** / Option 2*: **2005soc** | 2005soc | 2005soc |
| **III** | RCP6.0 climate & CO$_2$ | Climate & CO$_2$ | Experiment I | Experiment II | rcp60 | not simulated |
| | varying society & protection up to 2005, then fixed at 2005 levels thereafter | Human & LU | | | 2005soc | |
| **IV** | no climate change, pre-industrial CO$_2$ | Climate & CO$_2$ | Experiment I | Experiment I | picontrol | picontrol |
| | varying society & protection up to 2100 (SSP2), then fixed at 2100 levels thereafter | Human & LU | | | ssp2soc | 2100ssp2soc |
| **VI** | RCP2.6 climate & CO$_2$ | Climate & CO$_2$ | Experiment I | Experiment II | rcp26 | rcp26 |
| | varying society & protection up to 2100 (SSP2), then fixed at 2100 levels thereafter | Human & LU | | | ssp2soc | 2100ssp2soc |





| VII | RCP6.0 climate & $CO_2$ | Climate & $CO_2$ | Experiment I | Experiment II | rcp60 | not simulated |
|---|---|---|---|---|---|---|
| | varying society & protection (SSP2) | Human & LU | | | ssp2soc | |

### 8.10    Fisheries and Marine Ecosystems

The fisheries and marine ecosystem models are quite diverse. Most include climate-impact models via ESM-simulated primary-production changes, and many also include impacts of changes in water temperature on

5    ectotherm metabolic rates. A very small subset of the models includes ocean-acidification effects. Most models include fishing, either as an imposed process based on observed historical fishing effort (which start in 1950), or as an endogenous process based on simple economic factors.

Fishing effort should be held at constant 1950 levels from 1861-1950. It should then follow the standard historical reconstruction from 1950-2006 typically used by the model, using reconstructed effort or economic forcings as

10    appropriate. Effective effort should be held constant following 2005 in all simulations. For models that include acidification effects, all simulations should include ocean acidification in accordance with the respective climate scenario.

| Climate scenarios | |
|---|---|
| picontrol | Pre-industrial climate |
| historical | Historical climate and $CO_2$ concentration. |
| rcp26 | Future climate and $CO_2$ concentration from RCP2.6 |
| rcp60 | Future climate and $CO_2$ concentration from RCP6.0 |
| Human influences scenarios | |
| nosoc | No fishing |
| histsoc | Historical reconstruction of fishing starting in 1950 |
| 2005soc | Fishing fixed at year 2005 levels |





**Table 10** ISIMIP2b scenarios for simulations of the impacts on marine ecosystems and fisheries.

| | Experiment | Input | Pre-industrial 1661-1860 | Historical 1861-2005 | Future 2006-2100 | Extended future 2101-2299 |
|---|---|---|---|---|---|---|
| **I** | no climate change, pre-industrial $CO_2$ | Climate & $CO_2$ | **picontrol** | **picontrol** | **picontrol** | **picontrol** |
| | varying fishing up to 2005, then fixed at 2005 levels thereafter | Human & LU | **nosoc** | **histsoc** | **2005soc** | **2005soc** |
| **II** | RCP2.6 climate & $CO_2$ | Climate & $CO_2$ | Experiment I | **historical** | **rcp26** | **rcp26** |
| | varying fishing up to 2005, then fixed at 2005 levels thereafter | Human & LU | | **histsoc** | **2005soc** | **2005soc** |
| **III** | RCP6.0 climate & $CO_2$ | Climate & $CO_2$ | Experiment I | Experiment II | **rcp60** | not simulated |
| | varying fishing up to 2005, then fixed at 2005 levels thereafter | Human & LU | | | **2005soc** | |

## 8.11 Tropical cyclones

The occurrence of tropical cyclones is only influenced by climate change and independent of other human
influences. Therefore scenarios only depend on the climate input.

To simulate tropical cyclones, we use the downscaling technique described in detail by (Emanuel et al., 2008).
Broadly, the technique begins by randomly seeding with weak proto-cyclones the large-scale, time-evolving state
given by the CMIP5 climate model data. These seed disturbances are assumed to move with the GCM-provided
large-scale flow in which they are embedded, plus a westward and poleward component owing to planetary
curvature and rotation. Their intensity is calculated using the Coupled Hurricane Intensity Prediction System
(CHIPS; Emanuel et al., 2004), a simple axisymmetric hurricane model coupled to a reduced upper ocean model to
account for the effects of upper ocean mixing of cold water to the surface. Applied to the synthetically generated
tracks, this model predicts that a large majority of them dissipate owing to unfavorable environments. Only the
15 'fittest' storms survive; thus the technique relies on a kind of natural selection. Extensive comparisons to
historical events by Emanuel et al. (2008) and subsequent papers provide confidence that the statistical





properties of the simulated events are in line with those of historical tropical cyclones. We simulate 300 events globally each year and for each CMIP5 model, for the period 1950-2005 for the historical period, and 2006-2100 in downscaling the RCP2.6 and 6.0 cases, yielding a total of 16,800 simulated tropical cyclones for each model in the historical period, and 28,500 simulated cyclones per model for the RCP2.6 and 6.0 cases. The response to

global warming of both the frequency and intensity of the synthetic events compares favorably to that of more standard downscaling methods applied to the Coupled Model Intercomparison Project 3 (CMIP3) generation of climate models (Christensen et al., 2013).

| Climate & CO$_2$ scenarios | |
|---|---|
| **picontrol** | Pre-industrial climate |
| **historical** | Historical climate |
| **rcp26** | Future climate from RCP2.6. |
| **rcp60** | Future climate from RCP6.0. |

**Table 11** ISIMIP2b scenarios for tropical cyclone simulations.

| | Experiment | Input | Pre-industrial 1661-1860 | Historical 1861-2005 | Future 2006-2100 | Extended future 2101-2299 |
|---|---|---|---|---|---|---|
| **I** | no climate change | Climate | **picontrol** | not simulated | not simulated | not simulated |
| **II** | RCP2.6 climate | Climate | Experiment I | **historical** | **rcp26** | **rcp26** |
| **III** | RCP6.0 climate | Climate | Experiment I | Experiment II | **rcp60** | not simulated |

## 9   Intended time line of the simulations

The time line of ISIMIP2b has been chosen to meet the critical deadlines of the drafting process of the IPCC Special Report, with the submission deadline for papers to be considered in the Special Report being in October 2017 and the associated acceptance deadline being in April 2018. ISIMIP2b simulations are therefore envisaged to be completed well before October 2017. Ideally simulations for the first GCM should be completed by end of

2016 to leave enough time for the analysis of the data. Bias-corrected data for the IPSL-CM5A-LR and GFDL-ESM2M as well as the required patterns of sea level rise, LU and irrigation, as well as the other data sets described in Table 2 will soon become available. The additional climate input from HadGEM2-ES and MIROC5 is planned to be provided by end of 2016. The ISIMIP2b repository will stay open for impacts simulations submitted





beyond October 2017, since the described simulations provide a basis for further research beyond the direct demands of the Special Report, including for the IPCC Sixth Assessment Report.

## 10    Discussion

Our protocol addresses a timely and important research gap that we have identified for developing a framework for assessing the impacts of 1.5°C and 2°C global warming on a multitude of different impact sectors. Whilst a number of studies have investigated the impacts of 1.5°C and/or 2°C on individual impact sectors (Gosling et al., 2016; NW et al., 2014; Roudier P, Andersson JM, Donnelly C, Feyen L, Greuell W, 2015), our approach provides a novel extension to these by: 1) incorporating multiple GCMs, impact models and sectors; 2) providing a consistent and documented framework for the assessment of impacts at the global scale; and 3) seeking to achieve multi-model integration between sectors in order to better represent the links and feedbacks that occur in the observed Earth system.

The last novelty above, in particular, is a significant step-change in how climate-change-impact modelling is conducted, since up until now the assessment of global-scale climate-sensitive impacts for different sectors have typically been conducted in isolation of one another, e.g. the water-sector models do not use LU changes from the biomes-sector models, and in turn the crop-sector models do not use runoff from the water-sector models etc. Running impact models in isolation of one another can ignore complex interdependencies which in turn can be detrimental to the representation of spatial patterns in climate change impacts, as well as their sign and magnitude of change (Paula A. Harrison, Robert W. Dunford, 2016). Enhancing cross-sectoral integration has been one of the driving forces behind the development of the ISIMIP2b protocol, so we anticipate that the simulations which arise from it will yield some of the most cutting-edge projections of climate change impacts to date.

As well as facilitating an understanding of the impacts of 1.5°C and 2°C warming, the ISIMIP2b scenario design also enables an assessment of the impacts of the 1°C of global warming that has occurred between pre-industrial times and the present-day. There are surprisingly few studies that have investigated this, in part due to the significant resources needed to conduct the lengthy climate and impact simulations that are required. To understand what effect anthropogenic climate change has had since pre-industrial times requires an understanding of the climate-change conditions that would prevail in the present-day in the absence of anthropogenic greenhouse gas emissions as well as an estimate of how climate-sensitive impacts have responded to human-induced LU change and land-management since pre-industrial times.



To disentangle the magnitude of climate-sensitive impacts from changes in these impacts that have occurred due to other human activities, the scenario design compares a simulations, where human influences on climate-sensitive impacts occur under a pre-industrial climate, driven by stable greenhouse gas concentrations, with another simulation for the same time period, where the climate responds to increases in greenhouse gas emissions, and where there are human influences on climate-sensitive impacts. It seems intuitive that the difference between these two simulations will yield the pure effect of climate change, whilst controlling for human influences. However, we acknowledge that in practical terms, the effects of human activity on the climate, and climate-sensitive impacts respectively, are intrinsically linked and cannot be separated precisely. For example, whilst we are able to use historical estimates of water abstractions and dam construction as one of the human influences in both of the above simulations, a proportion of the abstractions and construction of dams will have occurred at the time in response to climate variability and based on decisions related to planning for future climate change. Such a caveat has to be accepted within the context of a numerical modelling framework such as ours.

However, the explicit representation of "other human influences" on impact indicators means an important step forward compared to the ISIMIP fast track simulations. In particular, the assessment of potential trade-offs of specific mitigation measures such expansion of bioenergy production will become critical when implementing the Paris agreement of limiting global warming to "well below 2°C".

**Code and/or data availability**

All input data described in Section 3 to Section 7 will be made publicly available. Availability will be announced on www.isimip.org where the way of accessing the data will also be described. Model output except for the hurricane projections will also be made publicly available via the ISIMIP repository https://esg.pik-potsdam.de. Access to the hurricane projections can be gained by request via info@windrisktech.com.

**Acknowledgements**

We thank Graham Weedon (Met Office) and Emanuel Dutra (ECMWF) who helped a lot to put together the EWEMBI dataset. COST Action FP1304 for supporting biomes meeting. This research was supported in part by the EU FP7 HELIX project (grant no. FP7-603864-2).



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
