# Peer review of "Assessing the impacts of 1.5°C global warming – simulation protocol of the Inter-Sectoral Impact Model Intercomparison Project (ISIMIP2b)"

_Geoscientific Model Development, 2016_

## Referee Comment (RC1) · A. J. Dolman (Referee) · 19 Jan 2017

This is a difficult paper to review. There is no doubt that it should be published, as it contains the deliberations that go in to developing scenarios to quantify the impacts of warming, socio economic trends and land use in an attempt to remain below the Paris goal of 1.5 C. The paper as such provides a very useful and needed reference for the impact community. The paper is also clear, contains no mistakes or errors.

That being said, the paper is very hard to read. My suggestion would be to keep the paper as it is up to section 8. In section 8, the level of detail, repeating of similar tables is such that anyone would loose direction. I appreciate that the sector specific implementation is important, but I would give only one example, and put the other

tables in the supplementary material.

On page 5, line 15-20 the item number 3 is missing.

---

## Short Comment (SC1) · 19 Jan 2017

Dear Dr. Dolman,

thanks a lot for your constructive review! We will consider how to make the paper more concise and readable during the revision; your suggestion to move some of the sector-specific material may indeed be a good starting point.

Best regards,

Jacob Schewe

---

## Referee Comment (RC2) · D. Jacob (Referee) · 14 Jun 2017

Dear All,

I suggest to publish the manuscript almost as it is. It is a good paper, well written. The method explained is solid and fits to the purpose.

There are a few minor issues:

the paper is very detailed. There is a lot of repetition of the simulations design. It is listed in many sections.

on page 4 in the middle paragraph it says that the data from ISIMPI2a will eventually

be publicly available. In order to help the community in this lind of studies and for trust building - they should be made publicly available. Otherwise it is not possible for outsiders to judge on the quality of all follow up and proposed simulation results.

At the end, the paper is too detailed and hard to read.

In chapter 3 it is unclear if the chosen GCMs are representative for a spread on possibilities. How do they compare to the full set of CMPI5 simulations.

What is the role of the bias correction (which by the way should be named bias- adjustment, since the bias will only be reduced but not corrected!!!) on the final results?

Since all impact models are so different and the input sets are also very different, I did not understand, how comparable the final results will be.

best regards

---

## Author Comment (AC1) · 28 Jul 2017

In the course of the ISIMIP2b simulation phase there have been some improvements of the protocol that do not directly refer to the reviewers' comments but have let to associated adjustments of the manuscript:

1. Instead of the originally suggested linear interpolation from historical land use patterns from HYDE3.2 to future projections generated by the LU model MAgPIE we have decided to apply a harmonization method that has recently been developed in the context of the CMIP6 process (Hurtt et al., in preparation). While the underlying MAgPIE simulations have not changed the main text and the SI have been adjusted to describe

the new harmonization approach. Figure 3 of the main text now shows the areas of crop land from the historical reconstruction and the original MAgPIE simulations without interpolations. The SI contains a comparison of the harmonized crop land areas and the original data for the IPSL-CM5A-LR climate model. The patterns for the other models are currently generated. George Hurtt, Louise Chini, Ritvik Sahapal, Benjamin Bodirsky, Jan Volkholz, and Steve Frolking have been added to the list of authors as they were involved in the generation of the new harmonized LUH2-ISIMIP2b land use data set.

2. We were able to integrate a section on the simulation of climate-change effects on lakes using coupled lake-hydrodynamic and water-quality models into the ISIMIP2b protocol. Rafael Marcé, Don Pierson and Jonas Jägermeyr have lead the development of the section and have been added to the list of co-authors.

3. Similarly, a "terrestrial biodiversity sector" has been added to the ISIMIP2b protocol. Christian Hof and Matthias Biber have worked on the development of the associated section of the protocol document (https://www.isimip.org/protocol/#isimip2b) together with Thomas Hickler and they have thus been added to the list of co-authors.

4. We now also provide spatially explicit GDP distributions as input for the ISIMIP2b simulations or to e.g. estimate damages in post-processing of bio-physical impact simulations. The new data set covers the period from 1860 to 2100 and is consistent with reported national GDP data for the historical period and future projections on national level following SSP2 (see section 6 of the main text). The development has been done by Daisuke Murakami, Yoshiki Yamagata, and Tobias Geiger who have been added to the list of co-authors.

5. We now propose a method to account for the effect of changes in Terrestrial Water Storage (TWS) e.g. due to projected changes in ground water abstraction according to SSP2 on sea level rise. The approach is designed to be consistent with projected changes in land use and irrigation patterns as provide within ISIMIP2b and will be

applied to generate spatially explicit patterns of sea level change for the Group 3 simulations within the coastal infrastructure sector. A description of the associated addition to the Group 2 sea level projections has been added to the paper. Riccardo Riva has been involved in the development of the approach and will contribute the sea level fingerprints associated with the projected changes in TWS. He has been added to the list of co-authors.

6. The hurricane simulations are no longer considered as an additional impact sector within ISIMIP2b but as a complement to the climate input data sets provided to force the impact models. Therefore the associated description has become part of Section 3 on climate input data.

Below we provide detailed answers to the review by A. J. Dolman:

This is a difficult paper to review. There is no doubt that it should be published, as it contains the deliberations that go in to developing scenarios to quantify the impacts of warming, socio economic trends and land use in an attempt to remain below the Paris goal of 1.5 C. The paper as such provides a very useful and needed reference for the impact community. The paper is also clear, contains no mistakes or errors.

Answer: We thank the reviewer for this generally positive judgement of our paper!

That being said, the paper is very hard to read. My suggestion would be to keep the paper as it is up to section 8. In section 8, the level of detail, repeating of similar tables is such that anyone would loose direction. I appreciate that the sector specific implementation is important, but I would give only one example, and put the other tables in the supplementary material.

Answer: We fully agree with the reviewer that the details listed in section 8 are more relevant for modelers who want to join ISIMIP2b than for general readers of the paper. We therefore generated a separate protocol document published on the ISIMIP website (https://www.isimip.org/protocol/#isimip2b) including all technical details required

to generate the impact models runs within the divers sectors, e.g. data formats and naming conventions. The document serves as the most-up to date reference for the modelers and now also includes the tables of scenarios and output data previously listed in section 8 and the SI. In this way 1) the paper becomes more readable and 2) we avoid confusion regarding the latest reference for technical details of the implementation that might have to be adjusted in the process of the project if they turned out to be impractical or confusing. We decided to only keep one table of scenarios in the main paper for illustration.

On page 5, line 15-20 the item number 3 is missing.

Answer: Thank you very much for the hint. We have adjusted the numbering.

―――――――――――――――――

---

## Author Comment (AC2) · 28 Jul 2017

In the course of the ISIMIP2b simulation phase there have been some improvements of the protocol that do not directly refer to the reviewers' comments but have let to associated adjustments of the manuscript:

1. Instead of the originally suggested linear interpolation from historical land use patterns from HYDE3.2 to future projections generated by the LU model MAgPIE we have decided to apply a harmonization method that has recently been developed in the context of the CMIP6 process (Hurtt et al., in preparation). While the underlying MAgPIE simulations have not changed the main text and the SI have been adjusted to describe

the new harmonization approach. Figure 3 of the main text now shows the areas of crop land from the historical reconstruction and the original MAgPIE simulations without interpolations. The SI contains a comparison of the harmonized crop land areas and the original data for the IPSL-CM5A-LR climate model. The patterns for the other models are currently generated. George Hurtt, Louise Chini, Ritvik Sahapal, Benjamin Bodirsky, Jan Volkholz, and Steve Frolking have been added to the list of authors as they were involved in the generation of the new harmonized LUH2-ISIMIP2b land use data set.

2. We were able to integrate a section on the simulation of climate-change effects on lakes using coupled lake-hydrodynamic and water-quality models into the ISIMIP2b protocol. Rafael Marcé, Don Pierson, and Jonas Jägermeyr have lead the development of the section and have been added to the list of co-authors.

3. Similarly, a "terrestrial biodiversity sector" has been added to the ISIMIP2b protocol. Christian Hof and Matthias Biber have worked on the development of the associated section of the protocol document (https://www.isimip.org/protocol/#isimip2b) together with Thomas Hickler and they have thus been added to the list of co-authors.

4. We now also provide spatially explicit GDP distributions as input for the ISIMIP2b simulations or to e.g. estimate damages in post-processing of bio-physical impact simulations. The new data set covers the period from 1860 to 2100 and is consistent with reported national GDP data for the historical period and future projections on national level following SSP2 (see section 6 of the main text). The development has been done by Daisuke Murakami, Yoshiki Yamagata, and Tobias Geiger who have been added to the list of co-authors.

5. We now propose a method to account for the effect of changes in Terrestrial Water Storage (TWS) e.g. due to projected changes in ground water abstraction according to SSP2 on sea level rise. The approach is designed to be consistent with projected changes in land use and irrigation patterns as provide within ISIMIP2b and will be

applied to generate spatially explicit patterns of sea level change for the Group 3 simulations within the coastal infrastructure sector. A description of the associated addition to the Group 2 sea level projections has been added to the paper. Riccardo Riva has been involved in the development of the approach and will contribute the sea level fingerprints associated with the projected changes in TWS. He has been added to the list of co-authors. 6. The hurricane simulations are no longer considered as an additional impact sector within ISIMIP2b but as a complement to the climate input data sets provided to force the impact models. Therefore the associated description has become part of Section 3 on climate input data.

Below we provide detailed answers to the review by D. Jacob:

1. Dear All, I suggest to publish the manuscript almost as it is. It is a good paper, well written. The method explained is solid and fits to the purpose.

Answer: Thank you very much for this positive evaluation of the paper!

2. There are a few minor issues: the paper is very detailed. There is a lot of repetition of the simulations design. It is listed in many sections.

Answer: To avoid repetition and increase the readability of the paper we have moved the largest part of the sector specific scenario lists from section 8 to the new ISIMIP2b protocol document including all technical details required to set-up the model simulations within the individual sectors. We also tried to avoid repetition of the scenario design across the other sections. In particular, the introduction has been re-structured and shortened.

3. on page 4 in the middle paragraph it says that the data from ISIMPI2a will eventually be publicly available. In order to help the community in this lind of studies and for trust building - they should be made publicly available. Otherwise it is not possible for outsiders to judge on the quality of all follow up and proposed simulation results.

Answer: Yes, it is the clear purpose of ISIMIP to make the impacts simulations of each

individual simulations round publically available. The quality checks of the ISIMIP2a data are nearly finished and a first package of the data is already listed on the ESGF server (https://esg.pik-potsdam.de) and will be released and freely available very soon. The others will follow soon after finalizing the doi assignment. For ISIMIP2b we even agreed with the modelers to forgo the usual "embargo period" of the archive and release the data as soon as possible to provide the opportunity to other researchers to work on the data and potentially provide additional input for the IPCC Special Report. Publication rules for the ISIMIP data in general and the ISIMIP2b output in particular are described in the "How to join ISIMIP" document (https://www.isimip.org/protocol/terms-of-use/) underlining the clear intention of providing open access to the simulation data comparable to e.g. the CMIP data. The associated reference is now included in the paper.

4. At the end, the paper is too detailed and hard to read.

Answer: We hope that we could solve this issue by moving large parts of the sector specific more technical information to the protocol document.

5. In chapter 3 it is unclear if the chosen GCMs are representative for a spread on possibilities. How do they compare to the full set of CMPI5 simulations.

Answer: To illustrate to what extent the 4 selected GCMs cover the range of projections provided by CMIP5 the SI now includes two additional plots. They show regional changes in annual temperatures and precipitation plotted against global mean temperature change for the considered 4 GCMs and the range of other models from CMIP5. As the bias-adjustment only preserves relative changes in precipitation on the grid level changes in the bias-adjusted input data cannot be directly compared to the non-bias-adjusted CMIP5 projections. Due to this problem, the comparison is based on the raw CMIP5 data. The selected regions cover the ISIMIP focus regions (river basins + oceanic regions) and global land and ocean temperatures. Regional changes are plotted against global mean temperature changes as ISIMIP2b is intended to quantify impacts at different levels of global warming, in particular the difference between impacts at 1.5°C and 2°C of global mean temperature change. In order to measure how well the ISIMIP2b set covers the ranges of regional climate change projections, we have added an analysis of the Fractional Range Coverage (FRC) as proposed by (McSweeney and Jones, 2016) to the SI. In sum the 4 GCMs (originally chosen on the basis of climate input data requirements) provide an FRC close to the mean FRC across randomly chosen four-member sets of CMIP5 GCMs.

6. What is the role of the bias correction (which by the way should be named bias-adjustment, since the bias will only be reduced but not corrected!!!) on the final results?

Answer: We have adjusted the naming throughout the manuscript. The bias-adjustment is critical for the impact simulations as the represented processes may show a non-linear response to changes in temperatures or other variables. So e.g. US crop yields have been shown to drop strongly as soon as temperature exceed a critical threshold of about 30°C, dams provide flood protections as long as water levels do not exceed a certain threshold, or mortality non-linearly depends on temperatures showing a strong increase beyond site-specific threshold. In many cases these temperature dependencies are implemented in absolute terms and biases in the input data could lead to a quite different historical distribution in impacts variables than the observed ones and changes in terms of global mean temperature are expected to depend on these starting conditions. To this end we decided in favor a bias-adjustment although it comes at the cost of losing full physical consistency of the climate variables.

7. Since all impact models are so different and the input sets are also very different, I did not understand, how comparable the final results will be.

Answer: ISIMIP is designed to allow for model intercomparison within sectors but also for an aggregation or integration of impacts across sectors. To this end climate and socio-economic inputs are harmonized and all modeling groups within a given sector are asked to provide the same list of output variables. Nevertheless, different decisions

about processes to implement (e.g. inclusion of direct human interferences or not), different ways of representing these processes, different parameter settings etc. may lead to strong differences between models. This is partly accounted for by considering different scenarios within the sectors (e.g. naturalized simulation (nosoc), simulations with constant present day management (pressoc) and simulations accounting for changes in management (varsoc)). So even models not accounting for the influences of direct (human) disturbances could be compared to others in a basic naturalized setting. In general ISIMIP is an ongoing process improving the mutual understanding of the models that may lead to improvements of the protocols and models in further rounds. In general, it has turned out that it is extremely helpful to start from a very basic intercomparison only building on a common climate input and identical socio-economic storylines avoiding too much harmonization.

---

## Author Response (AR2)

Dear Katja Frieler (and all),

Thanks for the revised version of the manuscript, which I evaluate as near ready for publication. I have only one concern for the completeness of the manuscript with the removal of the complete technical information to the ISMIP website. Could you update the SI so as to provide it there as well (see comments in details below).

Best wishes,
Didier (R.)

Non-public comments to the Author:
Dear Katja Frieler (and all),

I have read the response for reviewers and the revised manuscript. I am mostly happy with it. However I see one issue with the fact that a good part of the scenario description has been moved to a document on the ISMIP2 servers and that is the permanence of the link and the completeness of the manuscript. Though clearly the splitting that you did is going in the right way, could you either have a doi attributed to that document or (better I would think) have it added to the SI of the manuscript "as is". In that way the manuscript is still complete and the accepted version of the manuscript has its technical document attached to it.
Let me know what you think about this (by email if you wish to discuss it).
Best wishes,
Didier (R.)

Answer:

Dear Didier, thank you very much for the positive evaluation of our paper. We fully agree to your point that the paper should be stand-alone without a reference to a potentially changing online document. To this end we have added the scenario tables describing the model runs for each individual sector to the SI. In this way the paper includes all relevant information about the different settings. It is only the very technical information about file formats, variable names, or naming conventions for the output files that is still only included in the online document. These details may be adjusted in course of the project (e.g. an additional variable may be added to the list of output variables or we may still notice that a certain file format is more appropriate etc.). As we would like to avoid a situation where multiple versions of this technical information are available at different places, we decided to make them available in only one "living" online document. However, this information is not at all needed to understand the simulation setting itself and the reasoning behind it. In this sense we consider the paper stand-alone now.

There are a few more minor changes of the text based on adjustments that have been made in course of the project. The main modification refers to the description of the LU data provided within ISIMIP2b. While the harmonized data was only available for one GCM when we submitted the first revised version of the manuscript, the data is now available for all four GCMs. Therefore Figure 3 of the main text and Figure S6 of the SI have been adjusted accordingly. In addition we modified the approach to separate the bioenergy areas from the land use categories (c3per and c4per) provided by the harmonization approach by Hurtt et al.. While we first intended to use the information about the bioenergy fraction to total crop land from the original MAgPIE simulations, we finally decided to directly use the bioenergy fraction of c3per and c4per instead. This approach is actually more straight forward and transparent. It avoids a number of technical problems that arose from the previous method.

In addition, the paper now contains a full description of the bias-adjustment for Surface Downwelling Shortwave Radiation (rsds) and Near-Surface Relative Humidity (hurs). While we originally planned to describe the approach in a separate paper we decided to add it to the protocol paper in order to provide a full reference for the method without further delay.

Finally, after the submission of the first revised version of the manuscript, an rcp26soc and an rcp60soc scenario have been specified for the simulations in the regional forest sector. The associated descriptions have been added to table 6 of the main text.

We hope that the updated manuscript now fulfills all standards for its final publication.

Kind regards

Katja

[revised manuscript text omitted]

uncertainty range between the 5th and 95th percentile of the distribution. Blue: RCP2.6, yellow: RCP6.0. All timeseries relative to year 2005.

[Figure]

**Figure S3** Time series of Antarctica's combined contribution from solid ice discharge and surface mass balance changes to sea level rise based on global mean temperature change from GFDL-ESM2M (panel 1), HadGEM2-ES (panel 2), IPSL-CM5A-LR (panel 3) and MIROC5 (panel 4). Solid lines: Median projections, shaded areas: uncertainty range between the 5[th] and 95[th] percentile of the distribution. Blue: RCP2.6, yellow: RCP6.0. All timeseries relative to year 2005.

[Figure]

**Figure S4** Time series of the contribution of mountain glaciers to sea level rise based on global mean temperature change from GFDL-ESM2M (panel 1), HadGEM2-ES (panel 2), IPSL-CM5A-LR (panel 3) and MIROC5 (panel 4). Solid lines: Median projections, shaded areas: uncertainty range between the 5$^{th}$ and 95$^{th}$ percentile of the distribution. Blue: RCP2.6, yellow: RCP6.0. We here show the combined non-anthropogenic and anthropogenic glacier response. All timeseries relative to year 2005.

[Figure]

[Figure]

**Figure S5** The sea level contribution from glaciers is partly due to an ongoing adjustment to climatic changes before the time of human intervention. We here parametrize this adjustment to past natural climate variation by fitting a quadratic curve (black line) to the modeled non-anthropogenic of (Marzeion et al., 2014). In our parametrization the non-anthropogenic sea level rise from glaciers ceases in year 2056 and is assumed zero thereafter. Parametrized natural contribution from global glaciers: Blue line as in Marzeion et al., 2014; black line: quadratic fit).

**2 Comparison of MAgPIE crop land areas to the associated areas after harmonization**

[Figure]

[Figure]

**Figure S6: Comparison of areas of crop land from the original MAgPIE simulations and the harmonized version as derived for the four different climate projections.** Left-hand column: Historical reconstruction (black lines) + crop land areas associated with RCP2.6 (blue lines); right-hand column: Historical reconstruction (black lines) + crop land areas associated with RCP6.0 (orange lines). Line types separate land cover classes: rainfed or irrigated food/feed crops and rainfed or irrigated bioenergy crops (see legend). Light hues indicate original MAgPIE simulations, dark hues indicate the total crop land extent after harmonization.

**3   Separation of bioenergy areas (first level of disaggregation)**

The original harmonization method provides land use data at 0.25° resolution where it does not designate bioenergy areas within the five crop classes (c3per, c4per, c3ann, c4ann, and c3nfx). Before separation bioenergy areas cell fraction below 0.001 are set to 0 and all data are aggregated to 0.5°, the common grid of input (and output) data considered within ISIMIP2b.

In the original MAgPIE setting bioenergy grass and bioenergy trees are distinguished and considered as part of c4per and c3per, respectively. In addition, in the MAgPIE simulations and the associated areas of bioenergy production are considered purely rainfed. To separate bioenergy areas from the harmonized c4per and c3per areas we apply the following procedure: To reflect these features we used the following procedure to separate bioenergy areas from the 5 crop classes:

**Separation of bioenergy grass land from c4per**

The disaggregation of c4per into food/feed crops (c4per_food/feed) and biofuel (c4per_bf) builds on the fractions of biofuel to total c4per crop land provided by MAgPIE. To this end the MAgPIE information about the fractions is linearly interpolated to provide data for each year while originally it comes at 10-year time steps. However, the harmonization allocates c4per to grid cells that do not contain any c4per in the original MAgPIE simulations. To Nnevertheless, to determine an associated bioenergy fraction, the available information about bioenergy fractions from MAgPIE (c4per_bf / c4per) has been further extrapolated. This is done by an averaging across the nearest neighbors, where the size of the averaging window is increased until at least 3 non-NA values are found. Finally, total c4per_bf and c4per_food/feed are split up into rainfed and irrigated areas according to the irrigation fractions of total c4per derived from the harmonization. Resulting crop types are c4per_irrigated_food/feed, c4per_rainfed_food/feed, c4per_irrigated_bf, c4per_rainfed_bf, which add up to c4per provided by the harmonization. The separation applied here generates irrigated bioenergy grass land that does not exist in the original MAgPIE patterns. However, this approach is chosen to preserve the total area of irrigated agricultural land.

**Separation of areas bioenergy grass land from c3per**

Harmonized c3per are split into rainfed and irrigated fractions according to the c3per irrigation fractions provided by the harmonization method. In the original MAgPIE simulations c3per areas are substantially smaller than the c3per area derived from the harmonization due to different associations of the sub-crop type "others". Therefore it is not reasonable to extrapolate sparse MAgPIE c3per information on bioenergy fractions to all grid cells containing c3per after the harmonization. As the original MAgPIE projections do not show any c3per bioenergy crops before 2050, we also assume there is no c3per bioenergy until 2050 in the harmonized data. Thereafter we allocate any expansion of c3per areas in the harmonized data after 2050 to be due to bioenergy trees. The irrigation fraction of c3per_food/feed and c3per_bf is equal to the irrigation fraction of total c3per provided by the harmonization.

**4 Derivation of crop-specific land-use and irrigation patterns (second level of disaggregation)**

The historical reconstruction and the harmonized MAgPIE future projections only provides information on land use at a  aggregated level while many of the hydrological or biomes models that account for land use changes offer a  specific representation of different crops and therefore also require a more detailed representation of land use patterns as input for their simulations. While LUH2 offers a disaggregation of the historical HYDE3.2 patterns into 5 crop classes ($C_3$ annual, $C_3$ nitrogen-fixing, $C_3$ perennial, $C_4$ annual, $C_4$ perennial) many models even need further disaggregation. To allow for an efficient use of the land use information for the historical and future period we provide a further disaggregation of the historical and future agricultural land use categories into the following individual crops

1) maize, 2) groundnut, 3) rapeseed, 4) soybeans, 5) sunflower, 6) rice, 7) sugarcane and crop classes

1) pulses, 2) temperate cereals (incl. wheat), 3) temperate roots, 4) tropical cereals, 5) tropical roots, 6) others annual, 7) others perennial, and 8) others N-fixing.

For all classes we also separate between rainfed and irrigated areas based on the irrigation fractions provided by the LUH2-ISIMIP2b dataset. The disaggregation from the LUH2 categories to the finer classes is based on the harvested areas of 175 crops provided by Monfreda et al. (2008) for the year 2000. The share

$$x_{i,j} = \frac{C_{i,j}}{C_i}$$

of a specific class $C_{i,j}$ (e.g. "maize") in the broader class $C_i$ (e.g. "C$_4$ annual") is assumed to stay constant. For grid cells that contain crop land in the LUH2-ISIMIP2b data while they are not covered by crop land in Monfreda data set we apply a fraction $x_{i,j}$ that is representative of the country average crop mix the grid cell belongs to.

**5    Regional climate change projections of ISIMIP2b GCMs compared to CMIP5 GCMs**

Regional mean temperature and precipitation changes versus global mean temperature change as projected by a range of CMIP5 GCMs including those selected for ISIMIP2b are shown in Figures S6 and S7. To allow for a direct comparison it is based on the raw data of the ISIMIP2b GCMs before the bias-adjustment. In order to assess how well the ISIMIP2b set covers these ranges of regional climate change projections, an analysis of the Fractional Range Coverage (FRC) as proposed by (McSweeney and Jones, 2016) is presented in Figure S8. Here, the FRC is calculated for the slopes of the linear fit lines depicted in Figures S6 and S7. We generated 500 random four-member subsets of the GCMs included in Figures S6 and S7. For each of these subsets and for the ISIMIP2b subset we then calculated all regional temperature and precipitation change FRCs. Then we determined the subset that yields the greatest/least mean value of these FRCs. Following the McSweeney and Jones (2016) terminology, this subset is called the Best/Worst Global Set. For the sake of completeness, we also determined the Best/Worst Regional Sets, which are a collection of the subsets that yield the greatest/least intra-regional mean FRCs.

[Figure]

**Figure S7** Regional versus global annual mean temperature change under RCP8.5 as simulated by the four ISIMIP2b GCMs (colored) and other CMIP5 GCMs (grey, see legend) for the 2006-2099 time period and all ISIMIP2b focus regions (see Figure 6 and Table 8). Temperature change is defined relative to the respective 2006-2028 mean value. Straight lines represent least-square zero-intercept linear fits to the annual data depicted as crosses.

[Figure]

**Figure S8** Regional annual mean precipitation change versus global annual mean temperature change under RCP8.5 as simulated by the four ISIMIP2b GCMs (colored) and other CMIP5 GCMs (grey, see legend) for the 2006-2099 time period and all ISIMIP2b focus regions (see Figure 6 and Table 8). All precipitation and temperature changes are defined relative to the corresponding 2006-2028 mean value. Straight lines represent least-square zero-intercept linear fits to the annual data depicted as crosses.

[Figure]

**Figure S9** Distribution of the Fractional Range Coverage (FRC) of the regional temperature and precipitation change signals depicted as straight lines in Figures S6 and S7 for various (collections of) four-member subsets of the CMIP5 GCMs listed in Figures S6 and S7. Please see the text for a definition of the different subsets. For each subset or collection of subsets, left and right box-whisker plots represent distributions of regional temperature and precipitation change FRCs, respectively. The upper and lower whiskers are the maximum and minimum FRC, respectively. The thin horizontal lines represent the three quartiles. The thick line in the background represents the mean FRC of all regional temperature and precipitation change signals.

**6    Sector-specific implementation of scenario design**
* * *
**6.1     Biomes**

**Table S2**: ISIMIP2b scenarios for the global biomes simulations.

| | Experiment | Input | **Pre-industrial** **1661-1860** | **Historical** **1861-2005** | **Future** **2006-2100** | **Extended future** |
|---|---|---|---|---|---|---|
| | | | | | | |

| | | | | | | 2101-2299 |
|---|---|---|---|---|---|---|
| **I** | no climate change, pre-industrial $CO_2$ | Climate & $CO_2$ | **picontrol** | **picontrol** | **picontrol** | **picontrol** |
| | varying LU & human influences up to 2005, then fixed at 2005 levels thereafter | Human & LU | **1860soc** | **histsoc** | **2005soc** | **2005soc** |
| **II** | RCP2.6 climate & $CO_2$ | Climate & $CO_2$ | Experiment I | **historical** | **rcp26** | **rcp26** |
| | varying LU & human influences up to 2005, then fixed at 2005 levels thereafter | Human & LU | | **histsoc** | **2005soc** | **2005soc** |
| **IIa** | RCP2.6 climate, $CO_2$ after 2005 fixed at 2005 levels | Climate & $CO_2$ | Experiment I | Experiment II | **rcp26, 2005co2** | **rcp26, 2005co2** |
| |  LU & human influences  fixed at 2005 level _after 2005_ | Human & LU | | | **2005soc** | **2005soc** |
| **III** | RCP6.0 climate & $CO_2$ | Climate & $CO_2$ | Experiment I | Experiment II | **rcp60** | not simulated |
| |  LU & human influences  fixed at 2005 levels _after_ | Human & LU | | | **2005soc** | |

| | | | | | | |
|---|---|---|---|---|---|---|
| | 2005  | | | | | |
| **IV** | no climate change, pre-industrial $CO_2$ | Climate & $CO_2$ | Experiment I | Experiment I | **picontrol** | **picontrol** |
| | varying human influences & LU up to 2100 (RCP2.6), then fixed at 2100 levels thereafter | Human & LU | | | **rcp26soc** | **2100rcp26soc** |
| **V** | no climate change, pre-industrial $CO_2$ | Climate & $CO_2$ | Experiment I | Experiment I | **picontrol** | not simulated |
| | varying human influences & LU (RCP6.0) | Human & LU | | | **rcp60soc** | |
| **VI** | RCP2.6 climate & $CO_2$ | Climate & $CO_2$ | Experiment I | Experiment II | **rcp26** | **rcp26** |
| | varying human influences & LU up to 2100 (RCP2.6), then fixed at 2100 levels thereafter | Human & LU | | | **rcp26soc** | **2100rcp26soc** |
| **VII** | RCP6.0 climate & $CO_2$ | Climate & $CO_2$ | Experiment I | Experiment II | **rcp60** | not simulated |
| | varying human influences & LU (RCP6.0) | Human & LU | | | **rcp60soc** | |

**6.2 Regional Forest**

**Table S23**: ISIMIP2b scenarios for the regional forestry simulations.

| | Experiment | Input | Pre-industrial 1661-1860 | Historical 1861-2005 | Future 2006-2100 | Extended future 2101-2299 |
|---|---|---|---|---|---|---|
| **I** | no climate change, pre-industrial $CO_2$ | Climate & $CO_2$ | not simulated | **picontrol** | **picontrol** | **picontrol** |
| | varying LU & human influences up to 2005, fixed present-day management (BAU) afterwards | Human & LU | | **histsoc** | **2005soc** | **2005soc** |
| **II** | RCP2.6 climate & $CO_2$ | Climate & $CO_2$ | not simulated | **historical** | **rcp26** | **rcp26** |
| | varying LU & human influences up to 2005, fixed present-day management (BAU) afterwards | Human & LU | | **histsoc** | **2005soc** | **2005soc** |
| **IIa** | RCP2.6 climate, $CO_2$ fixed after 2005 | Climate & $CO_2$ | not simulated | Experiment II | **rcp26, 2005co2** | **rcp26, 2005co2** |
| | fixed present-day management (BAU) after 2005 | Human & LU | | | **2005soc** | **2005soc** |

| | | | | | | |
|---|---|---|---|---|---|---|
| **III** | RCP6.0 climate & CO$_2$ | Climate & CO$_2$ | not simulated | Experiment II | **rcp60** | not simulated |
| | fixed present-day management  after 2005 | Human & LU | | | **2005soc** | |
| **IV<s>a</s>** | no climate change, pre-industrial CO$_2$ | Climate & CO$_2$ | not simulated | Experiment I | **picontrol** | **picontrol** |
| | varying management ((forest management for mitigation) | Human & LU | | | **rcp26socAMsoc** | **2100rcp26socAMsoc** |
| **V** | no climate change, pre-industrial CO$_2$ | Climate & CO$_2$ | not simulated | Experiment I | **picontrol** | |
| | varying management (forest management for adaptation) | Human & LU | | | **rcp60soc** | |
| **VI<s>a</s>** | RCP2.6 climate & CO$_2$ | Climate & CO$_2$ | not simulated | Experiment II | **rcp26** | **rcp26** |
| | varying management (forest management for mitigation) | Human & LU | | | **rcp26<s>AM</s>soc** | **2100rcp26A<s>M</s>soc** |
| **VII** | RCP6.0 climate & CO$_2$ | Climate & CO$_2$ | not simulated | Experiment II | **rcp60** | |

| | varying management (forest management for adaptation) | Human & LU | | | rcp60soc | |
|---|---|---|---|---|---|---|

The regional forest simulations as described above are carried out once using the ISIMIP2b climate of the grid cell in which the forest sites are located and once using locally bias-adjusted data based on locally observed meteorological data.

**6.3 Permafrost**

**Table S3:** ISIMIP2b scenario specification for the permafrost simulations.

| | Experiment | Input | Pre-industrial 1661-1860 | Historical 1861-2005 | Future 2006-2100 | Extended future 2101-2299 | Beyond 2299 |
|---|---|---|---|---|---|---|---|
| **I** | no climate change, pre-industrial $CO_2$ | Climate & $CO_2$ | **picontrol** | not simulated | not simulated | not simulated | not simulated |
| | no other human influences | Human & LU | **nosoc** | | | | |
| **II** | RCP2.6 climate & $CO_2$ | Climate & $CO_2$ | Experiment I | **historical** | **rcp26** | **rcp26** | **2299rcp26** |
| | no other human | Human & | | **nosoc** | **nosoc** | **nosoc** | **nosoc** |

| | | | | | | | |
|---|---|---|---|---|---|---|---|
| | influences | LU | | | | | |
| **IIa** | RCP6.0 climate, CO$_2$ varying until 2005, then fixed at 2005 levels thereafter | Climate & CO$_2$ | Experiment I | Experiment II | rcp26, 2005co2 | rcp26, 2005co2 | 2299rcp26, 2005co2 |
| | no other human influences | Human & LU | | | nosoc | nosoc | nosoc |
| **III** | RCP2.6 climate & CO$_2$ | Climate & CO$_2$ | Experiment I | Experiment II | rcp60 | not simulated | not simulated |
| | no other human influences | Human & LU | | | nosoc | | |

**6.4 Agriculture**

**Table S4:** ISIMIP2b scenarios for global crop simulations. *Option 2 only if option 1 not possible.

| | Experiment | Input | Pre-industrial 1661-1860 | Historical 1861-2005 | Future 2006-2100 | Extended future 2101-2299 |
|---|---|---|---|---|---|---|
| | | | | | | |

| | | | | | | |
|---|---|---|---|---|---|---|
| | no climate change, pre-industrial $CO_2$ | Climate & $CO_2$ | **picontrol** | **picontrol** | **picontrol** | **picontrol** |
| **I** | varying management until 2005, then fixed at 2005 levels thereafter | Human & LU | Option 1*: **1860soc** | Option 1*: **histsoc** | **2005soc** | **2005soc** |
| | management fixed at 2005 levels | | Option 2*: **2005soc** | Option 2*: **2005soc** | | |
| **II** | RCP2.6 climate & $CO_2$ | Climate & $CO_2$ | Experiment I | **historical** | **rcp26** | **rcp26** |
| | varying management until 2005, then fixed at 2005 levels thereafter | Human & LU | | Option 1*: **histsoc** | **2005soc** | **2005soc** |
| | management fixed at 2005 levels | | | Option 2*: **2005soc** | | |
| **IIa** | RCP2.6 climate, $CO_2$ after 2005 fixed at 2005 levels | Climate | Experiment I | Experiment II | **rcp26, 2005co2** | **rcp26, 2005co2** |
| | management fixed at 2005 levels | Human & LU | | | **2005soc** | **2005soc** |
| **III** | RCP6.0 climate & $CO_2$ | Climate & $CO_2$ | Experiment I | Experiment II | **rcp60** | not simulated |

| | | | | | | |
|---|---|---|---|---|---|---|
| | varying management until 2005, then fixed at 2005 levels thereafter | Human & LU | | | **2005soc** | |
| **IV** | no climate change, pre-industrial $CO_2$ | Climate & $CO_2$ | Experiment I | Experiment I | **picontrol** | **picontrol** |
| | varying management up to 2100 (RCP2.6), then fixed at 2100 levels thereafter | Human & LU | | | **rcp26soc** | **2100rcp26soc** |
| **V** | no climate change, pre-industrial $CO_2$ | Climate & $CO_2$ | Experiment I | Experiment II | **picontrol** | not simulated |
| | varying management (RCP6.0) | Human & LU | | | **rcp60soc** | |
| **VI** | RCP2.6 climate & $CO_2$ | Climate & $CO_2$ | Experiment I | Experiment II | **rcp26** | **rcp26** |
| | varying management up to 2100 (RCP2.6), then fixed at 2100 levels thereafter | Human & LU | | | **rcp26soc** | **2100rcp26soc** |
| **VII** | RCP6.0 climate & $CO_2$ | Climate & $CO_2$ | Experiment I | Experiment II | **rcp60** | |
| | varying management (RCP6.0) | Human & | | | **rcp26soc** | |

| | | | LU | | | |
|---|---|---|---|---|---|---|

**6.5 Energy**

**Table S5:** ISIMIP2b scenarios for the simulations within the energy sector. *Option 2 only if option 1 not possible.

| | Experiment | Input | Pre-industrial 1661-1860 | Historical 1861-2005 | Future 2006-2100 | Extended future 2101-2299 |
|---|---|---|---|---|---|---|
| **I** | no climate change, pre-industrial $CO_2$ | Climate & $CO_2$ | picontrol | picontrol | picontrol | picontrol |
| | varying society up to 2005, then fixed at 2005 levels thereafter | Human & LU | Option 1: 1860soc | Option 1: histsoc | 2005soc | 2005soc |
| | fixed 2005 socio-economic conditions | | Option 2*: 2005soc | Option 2*: 2005soc | | |
| **Ib** | no climate change, pre-industrial $CO_2$ | Climate & $CO_2$ | picontrol | picontrol | picontrol | picontrol |
| | varying society up to 2015, then fixed at 2015 levels thereafter | Human & LU | Option 1: 1860soc | Option 1: histsoc | 2015soc | 2015soc |
| | fixed 2015 socio-economic | | Option 2*: | Option 2*: | | |

| | | | | | | |
|---|---|---|---|---|---|---|
| | conditions | | **2015soc** | **2015soc** | | |
| **II** | RCP2.6 climate & $CO_2$ | Climate& $CO_2$ | Experiment I | historical | rcp26 | rcp26 |
| | varying society up to 2005, then fixed at 2005 levels thereafter | LU etc. | | Option 1: histsoc | 2005soc | 2005soc |
| | fixed 2005 socio-economic conditions | | | Option 2*: 2005soc | | |
| **IIb** | RCP2.6 climate & $CO_2$ | Climate & $CO_2$ | Experiment Ia | historical | rcp26 | rcp26 |
| | varying society up to 2015, then fixed at 2015 levels thereafter | Human & LU | | Option 1: histsoc | 2015soc | 2015soc |
| | fixed 2015 socio-economic conditions | | | Option 2*: 2015soc | | |
| **III** | RCP6.0 climate & $CO_2$ | Climate & $CO_2$ | Experiment I | Experiment II | rcp60 | not simulated |
| | varying society up to 2005, then fixed at 2005 levels thereafter | LU etc. | | | 2005soc | |
| **IIIb** | RCP6.0 climate & $CO_2$ | Climate & | Experiment | Experiment | Rcp60 | not simulated |

| Group | Description | Variable | Exp col 1 | Exp col 2 | Scenario 1 | Scenario 2 |
|---|---|---|---|---|---|---|
| | | $CO_2$ | Ia | IIa | | |
| | varying society up to 2015, then fixed at 2015 levels thereafter | Human & LU | | | 2015soc | |
| IV | no climate change, pre-industrial $CO_2$ | Climate & $CO_2$ | Experiment I | Experiment I | picontrol | picontrol |
| | varying society up to 2100 (SSP2+RCP2.6), then fixed at 2100 levels thereafter | LU etc. | | | rcp26soc | 2100rcp26soc |
| V | no climate change, pre-industrial $CO_2$ | Climate | Experiment I | Experiment II | picontrol | not simulated |
| | varying society up to 2100 (SSP2+RCP6.0), then fixed at 2100 levels thereafter | LU etc. | | | rcp60soc | |
| VI | RCP6.0 climate & $CO_2$ | Climate | Experiment I | Experiment II | rcp26 | rcp26 |
| | varying society up to 2100 (SSP2+RCP2.6), then fixed at 2100 levels thereafter | LU etc. | | | rcp26soc | 2100rcp26soc |
| VII | RCP6.0 climate & $CO_2$ | Climate | Experiment I | Experiment II | rcp60 | |

| | varying society up (SSP2+RCP6.0) | LU etc. | | | rcp26soc | |

**6.46.6 Temperature-Related Mortality**

**Table S6:** ISIMIP2b scenarios for temperature-related mortality simulations. Option 2* only if option 1 not possible.

| | Experiment | Input | Pre-industrial 1661-1860 | Historical 1861-2005 | Future 2006-2100 | Extended future 2101-2299 |
|---|---|---|---|---|---|---|
| | no climate change | Climate | picontrol | picontrol | picontrol | picontrol |
| **I** | varying society up to 2005, then fixed at 2005 levels thereafter, no adaptation | Human | Option 1: 1860soc | Option 1: histsoc | 2005soc | 2005soc |
| | society fixed at 2005 levels, no adaptation | | Option 2*: 2005soc | Option 2*: 2005soc | | |
| **II** | RCP2.6 climate | Climate | | historical | rcp26 | rcp26 |
| | varying society up to 2005, then fixed at 2005 levels thereafter, no adaptation | Human | Experiment I | Option 1*: histsoc | 2005soc | 2005soc |

| | | | Experiment I | Experiment II | | |
|---|---|---|---|---|---|---|
| | society fixed at 2005 levels, no adaptation | | | Option 2*: **2005soc** | | |
| **III** | RCP6.0 climate | Climate | Experiment I | Experiment II | **rcp60** | not simulated |
| | society fixed at 2005 levels, no adaptation | Human | | | **2005soc** | |
| **IV** | no climate change | Climate | Experiment I | Experiment II | **picontrol** | **picontrol** |
| | varying society (SSP2) up to 2100, then fixed at 2100 levels thereafter, no adaptation | Human | | | **ssp2soc** | **2100ssp2soc** |
| **V** | Not simulated | | | | | |
| **VI** | RCP2.6 climate | Climate | Experiment I | Experiment II | **rcp26** | **rcp26** |
| | varying society (SSP2) up to 2100, then fixed at 2100 levels thereafter, no adaptation | Human | | | **ssp2soc** | **2100ssp2soc** |

| | | | | | | |
|---|---|---|---|---|---|---|
| **VIa** | RCP2.6 climate | Climate | Experiment I | Experiment II | **rcp26** | not simulated |
| | varying society (SSP2) with adaptation | Human | | | **ssp2soc_adapt** | |
| **VII** | RCP6.0 climate | Climate | Experiment I | Experiment II | **rcp60** | not simulated |
| | varying society (SSP2), no adaptation | Human | | | **ssp2soc** | |
| **VIIa** | RCP6.0 climate | Climate | Experiment I | Experiment II | **rcp60** | not simulated |
| | varying society (SSP2), with adaptation | Human | | | **ssp2soc_adapt** | |

**6.7    Coastal infrastructure**

**Table S7:**  ISIMIP2b scenario specification for the simulations of impacts on coastal infrastructure.

| | Experiment | Input | Pre-industrial 1661-1860 | Historical 1861-2005 | Future 2006-2100 | Extended future 2101-2299 |
|---|---|---|---|---|---|---|
| **I** | no climate change, pre-industrial $CO_2$ | Climate & $CO_2$ | **picontrol** | **picontrol** | **picontrol** | **picontrol** |
| | varying society & | Human & LU | Option 1: | Option 1: | **2005soc** | **2005soc** |

| | | | | | | |
|---|---|---|---|---|---|---|
| | protection up to 2005, then fixed at 2005 levels thereafter | | | **1860soc** | **histsoc** | | |
| | society & protection fixed at 2005 levels | | | Option 2*: **2005soc** | Option 2*: **2005soc** | | |
| **II** | RCP2.6 climate & CO$_2$ | Climate & CO$_2$ | | | **historical** | **rcp26** | **rcp26** |
| | varying society & protection up to 2005, then fixed at 2005 levels thereafter | Human & LU | Experiment I | Option 1*: **histsoc** | **2005soc** | **2005soc** |
| | society & protection fixed at 2005 levels | | | Option 2*: **2005soc** | | |
| **III** | RCP6.0 climate & CO$_2$ | Climate & CO$_2$ | Experiment I | Experiment II | **rcp60** | not simulated |
| | society & protection fixed at 2005 levels after 2005 | Human & LU | | | **2005soc** | |
| **IV** | no climate change, pre-industrial CO$_2$ | Climate & CO$_2$ | Experiment I | Experiment I | **picontrol** | **picontrol** |

| | | | | | | |
|---|---|---|---|---|---|---|
| | varying society & protection up to 2100 (SSP2), then fixed at 2100 levels thereafter | Human & LU | | | **ssp2soc** | **2100ssp2soc** |
| **VI** | RCP2.6 climate & $CO_2$ | Climate & $CO_2$ | Experiment I | Experiment II | **rcp26** | **rcp26** |
| | varying society & protection up to 2100 (SSP2), then fixed at 2100 levels thereafter | Human & LU | | | **ssp2soc** | **2100ssp2soc** |
| **VII** | RCP6.0 climate & $CO_2$ | Climate & $CO_2$ | Experiment I | Experiment II | **rcp60** | not simulated |
| | varying society & protection (SSP2) | Human & LU | | | **ssp2soc** | |

**6.8 Marine Ecosystems and Fisheries**

**Table S8:** ISIMIP2b scenarios for simulations of the impacts on marine ecosystems and fisheries.

| | **Experiment** | **Input** | **Pre-industrial** 1661-1860 | **Historical** 1861-2005 | **Future** 2006-2100 | **Extended future** 2101-2299 |
|---|---|---|---|---|---|---|
| | | | | | | |

| | | Input | Pre-industrial 1660-1860 | Historical 1861-2005 | Future 2006-2099 | Extended future 2101-2299 |
|---|---|---|---|---|---|---|
| **I** | no climate change, pre-industrial $CO_2$ | Climate & $CO_2$ | **picontrol** | **picontrol** | **picontrol** | **picontrol** |
| | varying fishing up to 2005, then fixed at 2005 levels thereafter | Human & LU | **nosoc** | **histsoc** | **2005soc** | **2005soc** |
| **II** | RCP2.6 climate & $CO_2$ | Climate & $CO_2$ | Experiment I | **historical** | **rcp26** | **rcp26** |
| | varying fishing up to 2005, then fixed at 2005 levels thereafter | Human & LU | Experiment I | **histsoc** | **2005soc** | **2005soc** |
| **III** | RCP6.0 climate & $CO_2$ | Climate & $CO_2$ | Experiment I | Experiment II | **rcp60** | not simulated |
| | varying fishing up to 2005, then fixed at 2005 levels thereafter | Human & LU | Experiment I | Experiment II | **2005soc** | not simulated |

**6.9 Terrestrial Biodiversity**

**Table S9:** ISIMIP2b scenarios for simulations of the impacts on terrestrial biodiversity

| | Experiment | Input | Pre-industrial 1660-1860 | Historical 1861-2005[1] | Future 2006-2099[2] | Extended future 2101-2299[2] |
|---|---|---|---|---|---|---|
| **I** | pre-industrial | Climate & | **picontrol** | **picontrol** | **picontrol** | **picontrol** |

| | | | | | | |
|---|---|---|---|---|---|---|
| | climate | $CO_2$ | | | | |
| | no other human influences | Human & LU | nosoc | nosoc | nosoc | nosoc |
| **II** | RCP2.6 climate | Climate & $CO_2$ | Experiment I | historical | rcp26 | rcp26 |
| | no other human influences | Human & LU | | nosoc | nosoc | nosoc |
| **III** | RCP6.0 climate | Climate & $CO_2$ | Experiment I | Experiment II | rcp60 | not simulated |
| | no other human influences | Human & LU | | | nosoc | |